# Differentially Private Federated $k$-Means Clustering with Server-Side Data

**Jonathan Scott** [1]  **Christoph H. Lampert** [1]  **David Saulpic** [2]

## Abstract

Clustering is a cornerstone of data analysis that is particularly suited to identifying coherent subgroups or substructures in unlabeled data, as are generated continuously in large amounts these days. However, in many cases traditional clustering methods are not applicable, because data are increasingly being produced and stored in a distributed way, e.g. on edge devices, and privacy concerns prevent it from being transferred to a central server. To address this challenge, we present FedDP-KMeans, a new algorithm for $k$-means clustering that is fully-federated as well as differentially private. Our approach leverages (potentially small and out-of-distribution) server-side data to overcome the primary challenge of differentially private clustering methods: the need for a good initialization. Combining our initialization with a simple federated DP-Lloyds algorithm we obtain an algorithm that achieves excellent results on synthetic and real-world benchmark tasks. Our code can be found at https://github.com/jonnyascott/fed-dp-kmeans. We also provide a theoretical analysis of our method that provides bounds on the convergence speed and cluster identification success.

## 1. Introduction

Clustering has long been the technique of choice for understanding and identifying groups and structures in unlabeled data. Effective algorithms to cluster non-private centralized data have been around for decades (Lloyd, 1982; Shi & Malik, 2000; Ng et al., 2001). However, the major paradigm shift in how data are generated nowadays presents new challenges that often prevent the use of traditional methods. For instance, the proliferation of smart phones and other wearable devices, has led to large amounts of data being

---
[*]Equal contribution  [1]Institute of Science and Technology Austria (ISTA) [2]CNRS & Université Paris Cité, Paris, France. Correspondence to: Jonathan Scott <jscott@ist.ac.at>.

*Proceedings of the $42^{nd}$ International Conference on Machine Learning*, Vancouver, Canada. PMLR 267, 2025. Copyright 2025 by the author(s).

generated in a decentralized manner. Moreover, the nature of these devices means that the generated data are often highly sensitive to users and should remain private. While public data of the same kind usually exists, typically there is much less of it, and it does not follow the same data distribution as the private client data, meaning that it cannot be used to solve the clustering task directly.

These observations have triggered the development of techniques for learning from decentralized data, most popularly *federated learning (FL)* (McMahan et al., 2017). Originally proposed as an efficient means of training supervised models on data distributed over a large number of mobile devices (Hard et al., 2019), FL has become the de facto standard approach to distributed learning in a wide range of privacy-sensitive applications (Brisimi et al., 2018; Ramaswamy et al., 2019; Rieke et al., 2020; Kairouz et al., 2021). However, it has been observed that, on its own, FL is not sufficient to maintain the privacy of client data (Wang et al., 2019; Geiping et al., 2020; Boenisch et al., 2023). The reason is that information about the client data, or even some data items themselves, might be extractable from the learned model weights. This is obvious in the case of clustering: imagine a cluster emerges that consists of a single data point. Then, this data could be read off from the corresponding cluster center, even if FL was used for training. Therefore, in privacy-sensitive applications, it is essential to combine FL with other privacy preserving techniques. The most common among these is *differential privacy (DP)* (Dwork, 2006), which we introduce in Section 2. DP masks information about individual data points with carefully crafted noise. This can, however, lead to a reduction in the quality of results, called the privacy-utility trade-off.

Several methods have been proposed for clustering private data that are either federated, but not DP compatible, or which are DP but don't work in FL settings, see Section 6. In this paper we close this gap by introducing FedDP-KMeans, a fully federated and differentially private $k$-means clustering algorithm. Our main innovation is a new initialization method, FedDP-Init, that leverages server-side data to find good initial centers. Crucially, we do not require the server data to follow the same distribution as the client data, making FedDP-Init applicable to a wide range of practical FL scenarios. The initial centers serve as input to FedDP-Lloyds, a simple federated and differentially private

variant of Lloyds algorithm (Lloyd, 1982). As we expand upon in Section 2, a good initialization is critical to obtaining a good final clustering. While this is already true for non-private, centralized clustering, it is especially the case in the differentially private, federated setting, where we are further limited by privacy and communication constraints in the number of times we can access client data and thereby refine our initialization. We report on experiments for synthetic as well as real datasets in two settings: when we wish to preserve individual *data point privacy*, as is common for cross-silo federated learning settings (Li et al., 2020), and *client-level privacy*, as is typically used in cross-device learning settings (McMahan et al., 2017). In both cases, FedDP-KMeans achieves clearly better results than all baseline techniques. We also provide a theoretical analysis, proving that under standard assumptions for the analysis of clustering algorithms (Gaussian mixture data with well-separated components), the cluster centers found by FedDP-KMeans converge exponentially fast to the true component means and the ground truth clusters are identified after only logarithmically many steps.

To summarise, our main contributions are as follows:

- We propose a novel differentially private and federated initialization method that leverages small, out-of-distribution server-side data to generate high-quality initializations for federated $k$-means.

- We introduce the first fully federated and differentially private $k$-means algorithm by combining this initialization with a simple DP federated Lloyd's variant.

- We provide theoretical guarantees showing exponential convergence to ground truth clusters under standard assumptions.

- We conduct extensive empirical evaluations, demonstrating strong performance across data-point and user-level privacy settings on both synthetic and real federated data.

## 2. Background

$k$-**Means Clustering**    Given a set of data points, $P = (p_1, \ldots, p_n)$ and any $2 \leq k \leq n$, the goal of $k$-means clustering is to find *cluster centers*, $\nu_1, \ldots, \nu_k$ that minimize

$$\sum_{i=1}^{n} \min_{j=1,\ldots,k} \|p_i - \nu_j\|^2. \qquad (1)$$

The cluster centers induce a partition of the data points: a point $p$ belongs to cluster $j$, if $\|p - \nu_j\| \leq \|p - \nu_{j'}\|$ for all $j, j'$, with ties broken arbitrarily (but deterministically). It is well established that solving the $k$-means problem optimally is NP-hard in general (Dasgupta, 2008). However, efficient

approximate algorithms are available, the most popular being Lloyd's algorithm (Lloyd, 1982). Given an initial set of centers, it iteratively refines their positions until a local minimum of (1) is found. A characteristic property of Lloyd's algorithm is that the number of steps required to converge and the quality of the resulting solution depend strongly on the initialization: the most commonly used initialization is the $k$-means++ algorithm (Arthur & Vassilvitskii, 2007).

**Federated Learning**    Federated learning is a design principle for training a joint model from data that is stored in a decentralized way on local clients, without those clients ever having to share their data with anybody else. The computation is coordinated by a central *server* which typically employs an iterative protocol: first, the server sends intermediate model parameters to the clients. Then, the clients compute local updates based on their own data. Finally, the updates are *aggregated*, e.g. as their sum across clients, either by a trusted intermediate or using cryptographic protocols, such as multi-party computation (Bonawitz et al., 2016; Talwar et al., 2024). The server receives the aggregate and uses it to improve the current model, then it starts the next iteration. Although this framework enables better privacy, by keeping client data stored locally, each iteration incurs significant communication costs. Consequently, to make FL practical, it is important to design algorithms that require as few such iterations as possible.

While the primary focus of FL is on decentralized client data, the server itself can also possess data of its own, though usually far less than the clients in total and not of the same data distribution. Such a setting is in fact common in practice, where e.g. data from public sources, synthetically generated data, anonymized data, or data from some consenting clients is available to the server (Hard et al., 2019; Dimitriadis et al., 2020; Gao et al., 2022; Scott & Cahill, 2024).

**Differential Privacy (DP)**    DP is a mathematically rigorous framework for computing summary information about a dataset (for us, its cluster centers) in such a way that the privacy of individual data items is preserved. Formally, for any $\varepsilon, \delta > 0$, a (necessarily randomized) algorithm $\mathcal{A} : \mathcal{P} \to \mathcal{S}$ that takes as input a data collection $P \in \mathcal{P}$ and outputs some values in a space $\mathcal{S}$, is called $(\epsilon, \delta)$ *differentially private*, if it fulfills that for every $S \subset \mathcal{S}$

$$\Pr[\mathcal{A}(P) \in S] \leq e^{\varepsilon} \Pr[\mathcal{A}(P') \in S] + \delta, \qquad (2)$$

where $P$ and $P'$ are two arbitrary *neighboring* datasets. We consider two notions of *neighboring* in this work: for standard *data-point-level privacy*, two datasets are neighbors if they are identical except that one of them contains an additional element compared to the other. In the more restrictive *client-level privacy*, we think of two datasets as a collection of per-client contributions, and two datasets are neighbors if they are identical, except that all data points of one of the

individual client are missing in one of them. Condition (2) then ensures that no individual data item (a data point or a client's data set) can influence the algorithm output very much. As a consequence, from the output of the algorithm it is not possible to reliably infer if any specific data item occurred in the client data or not. An important property of DP is its *compositionality*: if algorithms $\mathcal{A}_1, \ldots, \mathcal{A}_t$ are DP with corresponding privacy parameters $(\varepsilon_1, \delta_1), \ldots, (\varepsilon_t, \delta_t)$, then any combination or concatenation of their outputs is DP at least with privacy parameters $(\sum_{s=1}^{t} \varepsilon_s, \sum_{s=1}^{t} \delta_s)$. In fact, stronger guarantees hold, which in addition allows trading off between $\varepsilon$ and $\delta$, see (Kairouz et al., 2015). These cannot, however, be easily stated in closed form. Due to compositionality, DP algorithms can be designed easily by designing individually private steps and composing them.

In this work, we use two mechanisms to make computational steps DP: The *Laplace mechanism* (Dwork et al., 2006) achieves $(\varepsilon, 0)$ privacy by adding Laplace-distributed noise with scale parameter $\frac{S}{\varepsilon}$ to the output of the computation. Here, $S$ is the *sensitivity* of the step, i.e. the maximal amount by which its output can change when operating on two neighboring datasets, measured by the $L^1$-distance. The *Gaussian mechanism* (Dwork & Roth, 2014) instead adds Gaussian noise of variance $\sigma_G^2(\varepsilon, \delta; S) = \frac{2 \log(1.25/\delta) S^2}{\varepsilon^2}$ to ensure $(\varepsilon, \delta)$-privacy Here, the sensitivity, $S$, is measured with respect to $L^2$-distance. The above formulas show that stronger privacy guarantees, i.e. a smaller *privacy budget* $(\varepsilon, \delta)$, require more noise to be added. This, however, might reduce the accuracy of the output. Additionally, the more processing steps there are that access private data, the smaller the privacy budget of each step has to be in order to not exceed an overall target budget. In combination, this causes a counter-intuitive trade-off for DP algorithms that does not exist in this form for ordinary algorithms: accessing the data more often, e.g. more rounds of Lloyd's algorithm, might lead to lower accuracy results, because the larger number of steps has to be compensated by more noise per step. Consequently, a careful analysis of the privacy-utility trade-off is crucial for DP algorithms. In general, however, algorithms are preferable that access the private data as rarely as possible. In the context of $k$-means clustering this means that one can only expect good results by avoiding having to run many iterations of Lloyd's. Consequently, a good initialization is crucial for achieving high accuracy.

## 3. Method

We assume a setting of $m$ clients. Each client, $j$, possesses a dataset, $P^j \in \mathbb{R}^{d \times n_j}$, where each column is a data point. The server also has some data, $Q$, which can be freely shared with the clients, but that is potentially small and *out-of-distribution* (i.e. not following the client data distribution). The goal is to determine a $k$-means clustering

of the joint clients' dataset $P := \bigcup_{j=1}^{m} P^j$ in a *federated* and *differentially private* way. We propose FedDP-KMeans, which solves this task in two stages. the first, FedDP-Init (Algorithm 1), is our main contribution: it constructs an initialization to $k$-means by exploiting server-side data. The second, FedDP-Lloyds (Algorithm 2), is a simple federated DP-Lloyds algorithm, which refines the initialization.

### 3.1. FedDP-Init

**Sketch:** FedDP-Init has three steps: **Step 1** computes a projection matrix onto the space spanned by the top $k$ singular vectors of the client data matrix $P$. **Step 2** projects the server data onto that subspace, and computes a weight for each server point $q$ that reflects how many client points have $q$ as their nearest neighbor. **Step 3** computes initial cluster centers in the original data space by first clustering the weighted server data in the projected space and then refining these centers by a step resembling one step of Lloyd's algorithm on the clients, but with the similarity computed in the projected space. To ensure the privacy of the client data all above computations are performed with sufficient amounts of additive noise, and the server only ever receives noised aggregates of the computed quantities across all clients. Consequently, FedDP-Init is differentially private and fully compatible with standard FL and secure aggregation setups, as described in Section 2.

Intuitively, the goal of Step 1 is to project the data onto a lower-dimensional subspace that preserves the important variance (i.e. distance between the means) but reduces the variance in nuisance direction (in particular the intra-cluster variance). This construction is common for clustering algorithm that strive for theoretical guarantees, and was popularized by Kumar & Kannan (2010). Our key novelty lies in Step 2 and 3: here, we exploit the server data, essentially turning it into a proxy dataset on which the server can operate without any privacy cost. After one more interaction with the clients, the resulting cluster centers are typically so close to the optimal ones, that only very few (sometimes none at all) steps of Lloyd's algorithm are required to refine them. Our theoretical analysis (Section 4) quantifies this effect: for suitably separated Gaussian Mixture data, the necessary number of steps to find the ground truth clusters is at most logarithmic in the total number of data points.

In the rest of this section, we describe the individual steps in more detail. For the sake of simpler exposition, we describe only the setting of data-point-level DP. However, only minor changes are needed for client-level DP, see Section 5. As private budget, we treat $\delta$ as fixed for all steps, and denote the individual budgets of the three steps as $\varepsilon_1$, $\varepsilon_2$ and $\varepsilon_3$. We provide recommendations how to set these values given an overall privacy budget in Appendix H.1.

**Algorithm details – Step 1:** The server aims to compute

**Algorithm 1** FedDP-Init

1: **Input:** Client data sets $P^1, \ldots, P^m$, # of clusters $k$, privacy parameters $\varepsilon_1, \varepsilon_2, \varepsilon_{3G}, \varepsilon_{3L}, \delta$

2: **Step 1:** *// Projection onto top $k$ singular vectors*
3: **for** client $j = 1, \ldots, m$ **do**
4:     Client $j$ computes outer product $P^j(P^j)^T$
5: **end for**
6: Server receives noisy aggregate $\widehat{PP^T} = \sum_{j=1}^{m} P^j(P^j)^T + \mathcal{N}_{d \times d}(0, \sigma^2(\varepsilon_1, \delta; \Delta^2))$
7: Server forms a projection matrix $\Pi$ from top $k$ eigenvectors of $\widehat{PP^T}$

8: **Step 2:** *// Determine importance weights*
9: **for** client $j = 1, \ldots, m$ **do**
10:     Client $j$ receives $\Pi$ and $\Pi Q$ from server
11:     **for** every point $q \in \Pi Q$ **do**
12:         Client $j$ computes weight $w_q(\Pi P^j) := \big|\{p \in \Pi P^j \mid \forall q' \in \Pi Q, \|p - q\| \leq \|p - q'\|\}\big|$
13:     **end for**
14: **end for**
15: Server receives noisy aggregate $\widehat{w_q(\Pi P)} = \sum_{j=1}^{m} w_q(\Pi P^j) + \mathrm{Lap}(0, \frac{1}{\varepsilon_2})$ for each $q \in \Pi Q$

16: **Step 3:** *// Cluster projected server points and initialize*
17: Server computes cluster centers $\xi_1, .., \xi_k$ by running $k$-means clustering of $\Pi Q$ with per-sample weights $\widehat{w_q(\Pi P)}$
18: **for** client $j = 1, \ldots, m$ **do**
19:     Client $j$ receives $\xi_1, .., \xi_k$ from server
20:     Client $j$ computes $S_r^j = \{p \in P^j : \forall s, \|\Pi p - \xi_r\| \leq \|\Pi p - \xi_s\|\}$, for $r = 1, \ldots, k$
21:     Client $j$ computes $m_r^j = \sum_{p \in S_r^j} p$ and $n_j^r = |S_r^j|$
22: **end for**
23: Server receives noisy aggregates $\widehat{m_r} = \sum_{j=1}^{m} m_r^j + \mathcal{N}_d(0, \sigma^2(\varepsilon_{3G}, \delta; \Delta))$ and $\widehat{n_r} = \sum_{j=1}^{m} n_r^j + \mathrm{Lap}(0, \frac{1}{\varepsilon_{3L}})$
24: Server computes initial centers $\nu_r = \widehat{m_r}/\widehat{n_r}$ for $r = 1, \ldots, k$

25: **Output:** Initial cluster centers $\nu_1, .., \nu_k$

---

the top $k$ eigenvectors of the clients' data outer product matrix $PP^T$. However, in the federated setup, it cannot do so directly because it does not have access to the matrix $P$. Instead, the algorithm exploits that the overall outer product matrix can be decomposed as the sum of the outer products of each client data matrix, i.e. $PP^T = \sum_{j=1}^{m} P^j(P^j)^T$. Therefore, each client can locally compute their outer product matrix and the server only receives their noisy across-client aggregate, $\widehat{PP^T}$. We ensure the privacy of this computation by the Gaussian mechanism. The associated sensitivity is the maximum squared norm of any single data point, which is upper bounded by the square of the dataset

radius, $\Delta$. Consequently, a noise variance of $\sigma_G^2(\varepsilon_1, \delta; \Delta^2)$ ensures $(\varepsilon_1, \delta)$-privacy, as shown by Dwork et al. (2014).

The remaining operations the server can perform noise-free: it computes the top $k$ eigenvectors of $\widehat{PP^T}$ and forms the matrix $\Pi \in \mathbb{R}^{d \times d}$ from them, which allows projecting to the $k$-dimensional subspace spanned by these vectors (which we call *data subspace*). The projection provides a data-adjusted way of reducing the dimension of data vectors from potentially large $d$ to the much smaller $k$. This is an important ingredient to our algorithm, because in low dimension typically less noise is required to ensure privacy. The lower dimension also helps keep the communication between server and client small. The dimension $k$ is chosen, because for sufficiently separated clusters, one can expect the subspace to align well with the subspace spanned by the cluster centers. In that case, the projection will preserve inter-cluster variance but reduce intra-cluster variance, which improves the signal-to-noise ratio of the data.

**Step 2:** Next, the server computes per-point weights for its own data such that it can serve as a proxy for the data of the clients. The server shares with the clients the computed projection matrix $\Pi$, and its own projected dataset $\Pi Q$. Each client uses $\Pi$ to project its own data to the data subspace. Then, it computes a weight for each server point $q \in \Pi Q$ as, $w_q(\Pi P^j) = \big|\{p \in \Pi P^j \mid \forall q' \in \Pi Q, \|p - q\| \leq \|p - q'\|\}\big|$, that is, the count of how many of the client's projected points are closer to $q$ than to any other $q' \in \Pi Q$, breaking ties arbitrarily. The weights are sent to the server in aggregated and noised form. As an unnormalized histogram over the client data, the point weight has $L^1$-sensitivity 1. Therefore, the Laplace mechanism with noise scale $1/\varepsilon_2$ makes this step $(\varepsilon_2, 0)$-DP. The noisy total weights, $\widehat{w_q(\Pi P)}$ for $q \in \Pi Q$, provide the server with a (noisy) estimate of how many client data points each of its data points represents. It then runs $k$-means clustering on its projected data $\Pi Q$, where each point $q$ receives weight $\widehat{w_q(\Pi P)}$ in the $k$-means cost function, to obtain centers $\xi_1, \ldots, \xi_k$ in the data subspace.

**Step 3:** In the final step the server constructs centers in the original space. For this, it sends the projected centers $\xi_1, \ldots, \xi_k$ to the clients. For each projected cluster center $\xi_r$, each client $j$ computes the set of all points $p \in P^j$ whose closest center in the projected space is $\xi_r$, i.e. $S_r^j := \{p \in P^j : \forall s, \|\Pi p - \xi_r\| \leq \|\Pi p - \xi_s\|\}$. For any $r$, the union of these sets across all clients would form a cluster in the client data. We want the mean vector of this to constitute the $r$-th initialization center. For this, each client $j$ computes the sum, and the number, of their points in each cluster, $m_r^j = \sum_{p \in S_r^j} p$, $n_r^j = |S_r^j|$. Aggregated across all clients one obtains the global sum and count of the points in each cluster: $m_r = \sum_{j=1}^{m} m_r^j$ and $n_r = \sum_{j=1}^{m} n_r^j$. To make this step private, we first split $\varepsilon_3 = \varepsilon_{3G} + \varepsilon_{3L}$. For $m_r^j$, which has $L^2$-sensitivity $\Delta$, we apply the Gaussian

mechanism with variance $\sigma^2(\varepsilon_{3G}, \delta; \Delta)$. For $n_r$, which has the $L^1$-sensitivity is 1, we use the Laplace mechanisms with scale $1/\varepsilon_{3L}$. This ensures $(\varepsilon_{3G}, \delta)$ and $(\varepsilon_{3L}, 0)$ privacy, respectively, and therefore (at least) $(\varepsilon, \delta)$ privacy overall for this step. Finally, the server uses the noisy estimates of the total sums and counts, $\widehat{m_r}$ and $\widehat{n_r}$, to compute approximate means $\nu_r = \widehat{m_r}/\widehat{n_r}$, and outputs these as initial centers.

## 3.2. FedDP-Lloyds

The second step of FedDP-KMeans is a variant of Lloyd's algorithm that we adapt to a private FL setting. The basic observation here is that a step of Lloyd's algorithm can be expressed only as summations and counts of data points. Consequently, all quantities that the server requires can be expressed as aggregates over client statistics which allows us to preserve user privacy with secure aggregation and DP. Specifically, assume that we are given initial centers $\nu_1^0, \ldots, \nu_k^0$, and a privacy budget $(\varepsilon_4, \delta_4)$, which we split as $\varepsilon_4 = \varepsilon_{4G} + \varepsilon_{4L}$. For rounds $t = 1, \ldots, T$, we repeat the following steps. The server sends the latest estimate of the centers to the clients. Each client $j$ computes, for $r = 1, \ldots, k$, $S_r^j := \{p \in P^j : \forall s, \|p - \nu_r^{t-1}\| \leq \|p - \nu_s^{t-1}\|\}$, the set of points whose closest center is $\nu_r^{t-1}$. Note that in contrast to the initialization, the distance is measured in the full data space here, not the data subspace. The remaining steps coincide with the end of Step 3 above. Each client $j$ computes the summations and counts of their points in each cluster: $m_r^j = \sum_{p \in S_r^j} p$ and $n_j^r = |S_r^j|$. These are aggregated to $m_r = \sum_{j=1}^m m_r^j$ and $n_r = \sum_{j=1}^m n_r^j$, and made private by the Gaussian mechanisms with variance $\sigma^2(\varepsilon_{4G}/T, \delta/T, \Delta)$ and the Laplacian mechanism with scale $T/\varepsilon_{4L}$, respectively. The server receives the noisy total sums and counts $\widehat{m_r}$ and $\widehat{n_r}$, and it updates its estimate of the centers as $\nu_r^t = \widehat{m_r}/\widehat{n_r}$. Overall, the composition property of DP ensures that FedDP-Lloyds is at least $(\varepsilon_4, \delta)$-private.

The choice of noise parameters ensure that the combination of our two algorithms is differentially private:

**Theorem 1.** *FedDP-KMeans followed with FedDP-Lloyds is $(\varepsilon, \delta)$-DP for $\varepsilon = \varepsilon_1 + \varepsilon_2 + \varepsilon_{3G} + \varepsilon_{3L} + \varepsilon_{4G} + \varepsilon_{4L}$*

## 4. Theoretical analysis

We analyze the theoretical properties of FedDP-KMeans in the standard setting of data from a $k$-component Gaussian mixture, i.e. the data $P$ is sampled from a distribution $\mathcal{D}(x) = \sum_{j=1}^k w_j \mathcal{G}_j(x)$ with means $\mu_j$, covariance matrix $\Sigma_j$ and cluster weight $w_j$. The data is partitioned arbitrarily among the clients, i.e. each clients data is not necessarily distributed according to $\mathcal{D}$ itself. We denote by $G_j$ the set of samples from the $j$-th component $\mathcal{G}_j$: the goal is to recover the clustering $G_1, ..., G_k$. The server data, $Q \subset \mathbb{R}^d$, can be small and not of the same distribution as $P$.

---

**Algorithm 2** FedDP-Lloyds

1: **Input:** Initial centers $\nu_1^0, \ldots, \nu_k^0$, $P$, steps $T$, privacy parameters $\varepsilon_G, \varepsilon_L, \delta$
2: **for** $t = 1, \ldots, T$ **do**
3:    **for** client $j = 1, \ldots, m$ **do**
4:       Client $j$ receives $\nu_1^{t-1}, \ldots, \nu_k^{t-1}$ from Server
5:       **for** $r = 1, \ldots, k$ **do**
6:          Client $j$ computes $S_r^j := \{p \in P^j : \forall s, \|p - \nu_r^{t-1}\| \leq \|p - \nu_s^{t-1}\|\}$
7:          Client $j$ computes $m_r^j = \sum_{p \in S_r^j} p$, $n_j^r = |S_r^j|$
8:       **end for**
9:    **end for**
10:    Server receives $\widehat{m_r} = \sum_{j=1}^m m_r^j + \mathcal{N}_d(0, T\Delta^2 \sigma^2(\varepsilon_G/T, \delta))$ and $\widehat{n_r} = \sum_{j=1}^m n_r^j + \mathrm{Lap}(0, \frac{T}{\varepsilon_L})$
11:    Server computes centers $\nu_r^t = \widehat{m_r}/\widehat{n_r}$, $r = 1, \ldots, k$
12: **end for**
13: **Output:** Final cluster centers $\nu_1^T, .., \nu_k^T$

---

Our main result is Theorem 3, which states that FedDP-KMeans successfully clusters such data, in the sense that the cluster centers it computes converge to the ground truth centers, i.e. the means of the Gaussian parameters, and the induced clustering becomes the ground truth one. In doing so, the algorithm respects data-point differential privacy. For this result to hold, a *separation condition* is required (Definition 2). It ensures that the ground truth cluster centers are separated far enough from each other to be identifiable. We first introduce and discuss the separation condition and then state the theorem. The proof is in Appendix E and F.

**Definition 2** (Separation Condition)**.** *For a constant c, a Gaussian mixture $\big((\mu_i, \Sigma_i, w_i)\big)_{i=1,\ldots,k}$ with n samples is called c-separated if*

$$\forall i \neq j, \|\mu_i - \mu_j\| \geq c\sqrt{\frac{k}{w_i}}\sigma_{max}\log(n),$$

*where $\sigma_{max}$ is the maximum variance of any Gaussian along any direction. For some large enough constant c fixed independently of the input, we say that the mixture is separated[1]*

Note that the dependency in $\log(n)$ is unavoidable, because with growing $n$ also the chance grows that outliers occur from the Gaussian distributions: assigning each data point to its nearest mean would not be identical to the ground truth clustering anymore.

To prove the main theorem, two additional assumptions on $P$ are required: (1) the diameter of the dataset is bounded by $\Delta := O\big(\frac{k\log^2(n)\sqrt{d}\sigma_{max}}{\varepsilon w_{min}}\big)$ – so that the noise added to

---

[1] The constant $c$ is determined by prior work: see Awasthi & Sheffet (2012)

compute a private SVD preserves enough signal.[2] (2): there is not too many server data, namely $|Q| \leq \frac{\varepsilon n k \sigma_{\max}^2}{\Delta^2}$. This ensures the noise added Step 2 is not overwhelming compared to the signal. Note that conditions (1) and (2) can always be enforced by two preprocessing steps, which we present as part of the proof in Appendix E. In practice, however, they are typically satisfied automatically, see Appendix G.2. This allows use of the algorithm directly as stated.

**Theorem 3.** *Suppose that the client dataset $P$ is generated from a separated Gaussian mixtures with $n \geq \zeta_1 \frac{k \log^3 n \sqrt{d} \sigma_{max}}{\varepsilon^2 w_{min}^2}$ samples, where $\zeta_1$ is a universal constant, and that $Q$ contains a least one sample from each component of the mixture. Then, there is a constant $\zeta_2$ such that, under assumptions (1) and (2), the centers $\nu_1, ..., \nu_k$ that are computed after $T$ steps of FedDP-Lloyds satisfy with high probability*

$$\|\mu_i - \nu_i\| \leq \zeta_2 \cdot \left( 2^{-T} \cdot \sqrt{\frac{n\sigma_{max}^2}{|G_i|}} + \frac{T\Delta \log(n)}{\varepsilon n w_{min}} \right). \quad (3)$$

*Furthermore, there is a constant $\zeta_3$ such that, after $\zeta_3 \log(n)$ rounds of communication, the clustering induced by $\nu_1, ..., \nu_k$ is the ground-truth clustering $G_1, ..., G_k$.*

Note that assumption (1) implies that $\frac{\Delta \log(n)}{\varepsilon w_{\min}}$ is negligible compared to $n$. That means, the estimated centers converge exponentially fast towards the ground truth.

## 5. Experiments

We now present our empirical evaluation of FedDP-KMeans, which we implemented using the `pfl-research` framework (Granqvist et al., 2024). Our code can be found at https://github.com/jonnyascott/fed-dp-kmeans.

To verify the broad applicability of our method we run experiments in both the setting of data-point-level privacy, see Section 5.1, and client-level privacy, see Section 5.2. The appropriate level of privacy in FL is typically determined by which data unit corresponds to a human. In *cross-silo* FL we typically have a smaller number of large clients, e.g. hospitals, with each data point corresponding to some individual, so data point-level privacy is appropriate. In *cross-device* FL, we typically have a large number of clients, where each client is a user device such as a smartphone, so client-level privacy is preferable. Our chosen evaluation datasets reflect these dynamics.

Appendix G contains additional details regarding the experimental evaluation and Appendix H contains additional

experiments and ablation studies. In particular, in H.1, we investigate how to set important hyperparameters of FedDP-KMeans, such as the privacy budgets of the individual steps. In H.2, we examine what happens when not all target clusters are present in the server data. Finally, in H.3, we discuss how to use existing methods for choosing $k$, based on the data, in the context of FedDP-KMeans.

**Baselines** As natural alternatives to FedDP-KMeans we consider different initializations of $k$-means combined with FedDP-Lloyds. Two baselines methods use the server data to produce initialization: *ServerKMeans++* runs $k$-means++ on the server data, while *ServerLloyds* runs a full $k$-means clustering of the server data. The baselines can be expected to work well when the server data is large and of the same distribution as the client data. This, however, is exactly the situation where the server data would suffice anyway, so any following FL would be wasteful. In the more realistic setting where the server data is small and/or out-of-distribution, the baselines might produce biased and therefore suboptimal results. As a third baseline, we include the *SpherePacking* initialization of (Su et al., 2017). This data-independent technique constructs initial centroids that are suitably spaced out and cover the data space, see Appendix G.3 for details. None of the above baselines use client data for initialization. Therefore, they consume none of their privacy budget for this step, leaving all of it for the subsequent FedDP-Lloyds. We also report results for two methods that do not actually adhere to the differentially-private federated paradigm. *k-FED* (Dennis et al., 2021) is the most popular federated $k$-means algorithm. As we will discuss in Section 6 it does not exploit server data and does not offer privacy guarantees. *Optimal* we call the method of transferring all client data to a central location and running non-private $k$-means clustering. This provides neither the guarantees of FL nor of DP, but is a lower bound on the $k$-means cost for all other methods.

**Evaluation Procedure** We compare FedDP-KMeans with the baselines over a range of privacy budgets. Specifically, if a method has $s$ steps that are $(\varepsilon_1, \delta), \ldots, (\varepsilon_s, \delta)$ DP then the total privacy cost of the method is computed as $(\varepsilon_{\text{total}}, \delta)$ by strong composition using Google's `dp_accounting` library [3]. We fix $\delta = 10^{-6}$ for all privacy costs. We vary the $\varepsilon_i$ of individual steps as well as other hyperparameters, e.g. the number of steps of FedDP-Lloyds, and measure the $k$-means cost of the final clustering. For each method we plot the Pareto front of the results in ($k$-means cost, $\varepsilon_{\text{total}}$) space. When plotting we scale the $k$-means cost by the dataset size, so the value computed in Equation 1 is scaled by $1/n$. This evaluation procedure gives us a good overview of the performance of each method at a range of privacy budgets. However, on its own it does not tell us how to set hyperparameters for FedDP-KMeans,

---

[2]It may be surprising to see that the diameter is allowed to increase when the privacy budget $\varepsilon$ gets smaller. However, Theorem 3 also requires the sample size to increase with $1/\varepsilon$, which counterbalances the growth of $\Delta$.

[3]https://github.com/google/differential-privacy/tree/main/python/dp_accounting

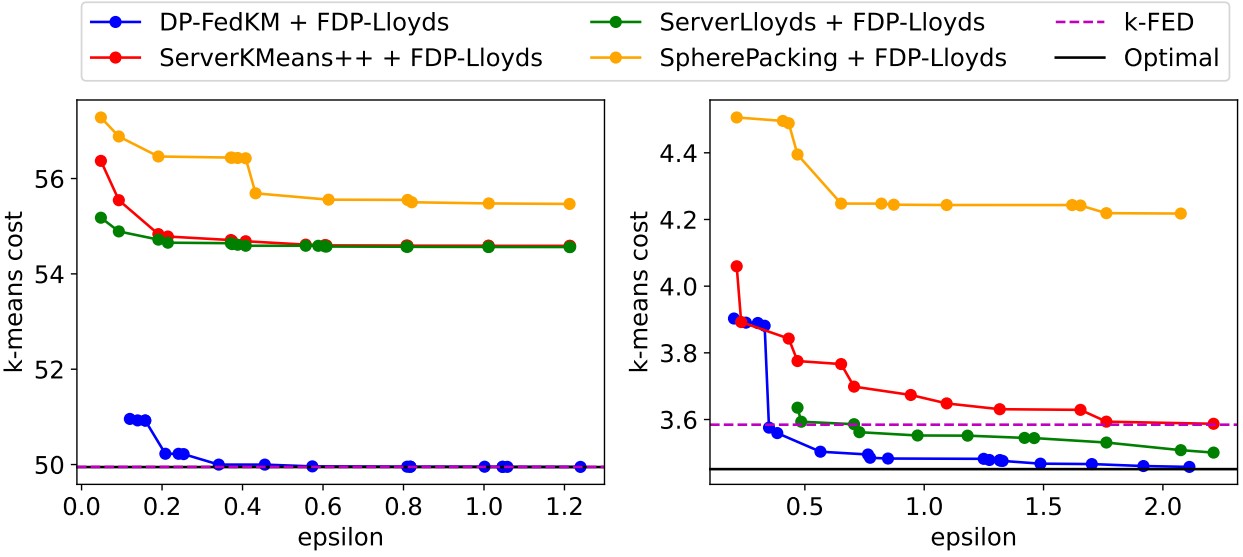

*Figure 1.* Results with data-point-level privacy ($k = 10$). Left: synthetic mixture of Gaussians data with 100 clients. Right: US census dataset. The 51 clients are US states, each client has the data of individuals with employment type "Federal government employee".

such how much privacy budget to assign to each step. Knowing how to do this is important for using FedDP-KMeans in practice and we address this in Appendix H.1.

### 5.1. Data-point-level Privacy Experiments

**Privacy Implementation details** In our theoretical discussion we assumed that no individual data point has norm larger than $\Delta$ in order to compute the sensitivity of certain steps. As $\Delta$ is typically not known in practice, in our experiments we ensure the desired sensitivity by clipping the norm of each data point to be at most $\Delta$, before using it in any computation. $\Delta$ is now a hyperparameter of the algorithm, which we set to be the radius of the server dataset.

**Datasets** We evaluate on synthetic and real federated datasets that resemble a cross-silo FL setting. Our synthetic data comes from a mixture of Gaussians, as assumed for our theoretical results in Section 4. The client data is of this mixture distribution. To simulate related, but OOD data, the server data consists of two thirds data from the true mixture and one third data that is uniformly distributed. We additionally evaluate on US census data using the folktables (Ding et al., 2021) package. The dataset has 51 clients, each corresponding to a US state. Each data point contains the information about a person in the census. We create a number of clustering tasks by filtering the client data to contain only those individuals with some chosen employment type. The server then recieves a small amount of data of individuals with a different employment type, to simulate related but OOD data. Full details on the datasets are in Appendix G.1.

**Results** See Figure 1. The left panel shows results for

the Gaussian mixture and the right panel for the US census dataset when the clients hold the data of federal employees. The other two categories are shown in Figures 8 and 9 of Appendix I. On synthetic data, FedDP-KMeans outperforms all DP baselines by a wide margin. These baselines cannot overcome their poor initializations, with performance plateauing even as the privacy budget grows. In contrast FedDP-KMeans obtains optimal (non-private) performance at a low privacy budget of around $\varepsilon_{\text{total}} = 0.4$. The non-private $k$-FED also performs optimally in this setting as is to be expected given that the synthetic data fulfills the assumptions of Dennis et al. (2021). On the US census datasets we observe a more interesting picture. Over all three settings FedDP-KMeans outperforms the baselines, except in the very low privacy budget regime. The latter is to be expected since for sufficiently low privacy budgets a client-based initialization will become very noisy, whereas the initialization with only server data (which requires no privacy budget) stays reasonable. With a high enough privacy budget FedDP-KMeans recovers the optimal non-private clustering. Among the baselines we observe similar performance between the two methods that initialize using server data, with ServerLloyds performing slightly better overall. The data independent SpherePacking performs poorly, emphasizing the importance of leveraging related server data to initialize. We attribute FedDP-KMeans's good performance predominantly to the excellent quality of its initialization. As evidence, Table 4 in Appendix G shows how many steps of Lloyd's had to be performed for Pareto-optimal behavior: this is never more than 2, and often none at all.

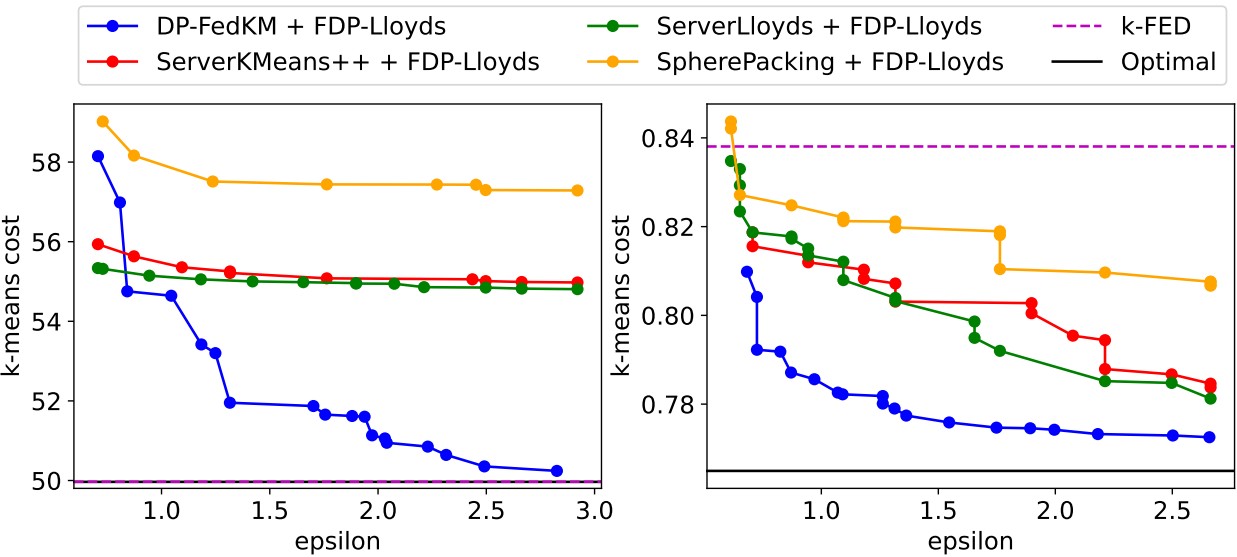

*Figure 2.* Results with client-level privacy ($k = 10$). Left: synthetic mixture of Gaussians data with 2000 clients. Right: stackoverflow dataset with 9237 clients, topic tags github and pdf.

## 5.2. Client-level Privacy Experiments

**Privacy Implementation details**    Moving to client-level differential privacy changes the sensitivities of the steps of our algorithms, which now depend not only on the maximum norm of a client data point norm, but also on the maximum number of data points a client has. Rather than placing assumptions on this, and deriving corresponding bounds on the sensitivity of each step, we instead simply enforce sensitivity by clipping client statistics prior to aggregation. This is a standard technique to enforce a given sensitivity in private FL, where it is typically applied to model/gradient updates. For full details on our implementation in the client-level privacy setting see Appendix G.4

**Datasets**    We evaluate on both synthetic and real federated datasets in a cross-device FL setting. For synthetic data we again use a mixture of Gaussians, but with more clients than in Section 5.1. We also use the Stack Overflow dataset from Tensorflow Federated. This is a large scale text dataset of questions posted by users on stackoverflow.com. We preprocess this dataset by embedding it with a pre-trainined sentence embedding model. Thus each client dataset consists of small number of text embedding vectors. The server data consists of embedding vectors from questions asked about different topics to the client data. See Appendix G.1.

**Results**    In Figure 2 we report the outcomes. The left shows results for the synthetic Gaussian mixture dataset with 2000 clients, and the right for the stackoverflow dataset, with topics github and pdf. Further results are in Appendix I: synthetic data with 1000 and 5000 clients in Figures 10 and 11, and the other stackoverflow topics are shown in

Figures 12, 13 and 14. For the synthetic data we observe that the baselines that use only server data are unable to overcome their poor initialization, even with more generous privacy budgets. As the total number of clients grows, from 1000 to 2000 to 5000, FedDP-KMeans exhibits better performance for the same privacy budget and the budget at which FedDP-KMeans outperforms server initialization becomes smaller. This is to be expected since the more clients we have the better our amplification by sub-sampling becomes. For stackoverflow we observe that FedDP-KMeans exhibits the best performance, except for in a few cases in the low privacy budget regime. $k$-FED performs quite poorly overall, tending to be outperformed by the private baselines. As in data-point-level privacy, we find the quality of FedDP-Init's initialization to be excellent: few, if any, Lloyd's steps are required for optimality (see Table 5, Appendix G).

## 6. Related Work

Within FL, clustering appears primarily for the purpose of grouping clients together. Such *clustered FL* techniques find a clustering of the clients while training a separate ML model on each cluster (Sattler et al., 2020; Ghosh et al., 2020; Xia et al., 2020). In contrast, in this work we are interested in the task of clustering the clients' data points, rather than the clients. In Dennis et al. (2021), the one-shot scheme $k$-Fed is proposed for this task: first all clients cluster their data locally. Then, they share their cluster centers with the server, which clusters the set of client centers to obtain a global clustering of the data. However, due to the absence of aggregation of the quantities that clients share

with the server, the method has no privacy guarantees. Liu et al. (2020) propose using *federated averaging* (McMahan et al., 2017) to minimize the $k$-means objective in combination with multi-party computation. Similarly, Mohassel et al. (2020) describe an efficient multi-party computation technique for distance computations. This will avoid the server seeing individual client contributions before aggregation, but the resulting clustering still exposes private information.

For private clustering, many methods have been proposed based on DPLloyd's (Blum et al., 2005), i.e. Lloyd's algorithm with noise added to intermediate steps. The methods differ typically in the data representation and initialization. For example, Su et al. (2016) creates and clusters a proxy dataset by binning the data space. This is, however, tractable only in very low dimensions. Ren et al. (2017) chooses initial centers by forming subsets of the original data and clustering those. Zhang et al. (2022) initializes with randomly selected data points and uses multi-party computation to securely aggregate client contributions. None of the methods are compatible with the FL setting. On the other hand, algorithms designed for Local Differential Privacy Chang et al. (2021); Dupré la Tour et al. (2024) could directly work in FL; albeit with an additive error so large that it makes any implementation impractical (Chaturvedi et al., 2022). DP algorithms that work in parallel environment, such as in (Cohen-Addad et al., 2022) could potentially be adapted to FL, however requiring a large number of communication rounds. Zhang et al. (2025) also work on federated clustering but with a focus on an asynchronous and heterogeneous setting rather than on differential privacy. To our knowledge, only two prior works combine the advantages of DP and FL. Li et al. (2023) is orthogonal to our work, as it targets *vertical FL* (all clients posses the same data points, but different subsets of the features). Diaa et al. (2024) studies the same problem as we do, but they propose a custom aggregation scheme that does not fit standard security requirements of FL. It uses SpherePacking to initialize, which in our experiments led to poor results.

## 7. Conclusion

In this paper we presented FedDP-KMeans, a federated and differentially private $k$-means clustering algorithm. FedDP-KMeans uses out-of-distribution server data to obtain a good initialization. Combined with a simple federated, DP, variant of Lloyd's algorithm we obtain an efficient and practical clustering algorithm. We show that FedDP-KMeans performs well in practice with both data-point-level and client-level privacy. FedDP-KMeans comes with theoretical guarantees showing exponential convergence to the true cluster centers in the Gaussian mixture setting. A shortcoming of our method is the need to choose hyperparameters, which is difficult when privacy is meant to be ensured. While we pro-

vide heuristics for this in Appendix H.1, a more principled solution would be preferable. It would also be interesting to explore if the server data could be replaced with a private mechanism based on client data.

## Acknowledgments

This research was funded in part by the Austrian Science Fund (FWF) [10.55776/COE12] and supported by the Scientific Service Units (SSU) of ISTA through resources provided by Scientific Computing (SciComp).

## Impact Statement

This paper presents work whose goal is to advance the field of Machine Learning. There are many potential societal consequences of our work, none which we feel must be specifically highlighted here.

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

# A. Extended related work

**Clustering Gaussian mixture**    The problem of clustering Gaussian mixtures is a fundamental of statistics, perhaps dating back from the work of Pearson (1894).

Estimating the parameters of the mixture, as we are trying to in this paper, has a rich history. Moitra & Valiant (2010) showed that, even non-privately, the sample complexity has to be exponential in $k$; the standard way to bypass this hardness is to require some separation between the means of the different components. If this separation is $o(\sqrt{\log k})$, then any algorithm still requires a non-polynomial number of samples (Regev & Vijayaraghavan, 2017). When the separation is just above this threshold, namely $O(\log(k)^{1/2+c})$, Liu & Li (2022) present a polynomial-time algorithm based on Sum-of-Squares to recover the means of *spherical* Gaussians.

For clustering general Gaussians, the historical approach is based solely on statistical properties of the data, and requires a separation $\Omega(\sqrt{k})$ times the maximal variance of each component (Achlioptas & McSherry, 2005; Awasthi & Sheffet, 2012). This separation is necessary for accurate clustering, namely, if one aims at determining from which component each samples is from (Diakonikolas et al., 2022). This approach has been implemented privately by Kamath et al. (2019) (with the additional assumption that the input is in a bounded area): this is the one we follow, as the simplicity of the algorithms allows to have efficient implementation in a Federated Learning environment. Bie et al. (2022) studied how public data can improve performances of this private algorithm: they assume access to a small set of samples from the distribution, which improves the sample complexity and allows them to remove the assumption that the input lies in a bounded area.

We note that both private works of Kamath et al. (2019) and Bie et al. (2022) have a separation condition that grows with $\log n$, as ours.

To only recover the means of the Gaussians, and not the full clustering, a separation of $k^\alpha$ (for any $\alpha > 0$) is enough (Hopkins & Li, 2018; Kothari et al., 2018; Steurer & Tiegel, 2021). This is also doable privately (when additionally the input has bounded diameter) using the approach of Cohen et al. (2021) and Tsfadia et al. (2022). Those works are hard to implement efficiently in our FL framework for two reasons: first, they rely on Sum-of-Square mechanisms, which does not appear easy to implement efficiently. Second, they use Single Linkage as a subroutine: this does not seem possible to implement in FL. Therefore, some new ideas would be necessary to get efficient algorithm for FL based on this approach.

A different and orthogonal way of approaching the problem of clustering Gaussian mixtures is to recover a distribution that is $\varepsilon$-close to the mixture in total variation distance, in which case the algorithm of Ashtiani et al. (2020) has optimal sample complexity $\tilde{O}(kd^2/\varepsilon^2)$ – albeit with a running time $\omega(\exp(kd^2))$.

**On private $k$-means clustering**    The private $k$-means algorithm of Dupré la Tour et al. (2024), implemented in our FL setting, would require either $\Omega(k)$ rounds of communication with the clients (for simulating their algorithm for central DP algorithm), or a a very large amount of additive noise $k^{O(1)}$ (for their local DP algorithm, with an unspecified exponent in $k$). Furthermore, the algorithm requires to compute a net of the underlying Euclidean space, which has size exponential in the dimension, and does not seem implementable. To the best of our knowledge, the state-of-the-art implementation of $k$-means clustering is from Chang & Kamath (2021): however, it has no theoretical guarantee, and is not tailored to FL.

# B. Technical preliminaries

### B.1. Differential Privacy definitions and basics

As mentioned in introduction, one of the most important properties of Differential Privacy is the ability to compose mechanisms. There are two ways of doing so. First, parallel composition: if an $(\varepsilon, \delta)$-DP algorithms is applied on two distinct datasets, then the union of the two output is also $(\varepsilon, \delta)$-DP. Formally, the mechanism that takes as input two elements $P_1, P_2 \in \mathcal{P}$ and outputs $(\mathcal{A}(P_1), \mathcal{A}(P_2))$ is $(\varepsilon, \delta)$-DP.

The second property is sequential composition: applying an $(\varepsilon, \delta)$-DP algorithm to the output of another $(\varepsilon, \delta)$-DP algorithm is $(2\varepsilon, 2\delta)$-DP. Formally: if $\mathcal{A} : \mathcal{P} \to \mathcal{S}_A$ is $(\varepsilon_A, \delta_A)$-DP and $\mathcal{B} : \mathcal{P} \times \mathcal{S}_A \to \mathcal{S}_B$ is $(\varepsilon_B, \delta_B)$-DP, then $\mathcal{B}(\mathcal{A}(\cdot), \cdot) : \mathcal{P} \to \mathcal{S}_B$ is $(\varepsilon_A + \varepsilon_B, \delta_A + \delta_B)$-DP.

Those are the composition theorem that we use for the theoretical analysis. We chose for simplicity not to use advanced composition: the number of steps in our final algorithms Algorithm 4 and Algorithm 5 is logarithmic, hence the improvement would be marginal. However, in practice, better bounds can be computed – although they don't have closed-form expression. We use a standard algorithm to estimate more precise upper-bounds on the privacy parameters of our algorithms (Kairouz

et al., 2015).

The sensitivity of a function is a key element to know how much noise is needed to add in order to make the function DP. Informally, the sensitivity measures how much the function can change between two neighboring datasets. Formally, we have the following definition.

**Definition 4** (Sensitivity). *Given a norm $\ell : \mathbb{R}^d \to \mathbb{R}$, the $\ell$-sensitivity of a function $f : \mathcal{X}^n \to \mathbb{R}^d$ is defined as*

$$\sup_{x \sim X' \in \mathcal{X}^n} \ell(f(X) - f(X')),$$

*where $X \sim X'$ means that $X$ and $X'$ are neighboring datasets.*

The two most basic private mechanism are the Laplace and Gaussian mechanism, that make a query private by adding a simple noise. We use the Laplace mechanism for counting:

**Lemma 5** (Laplace Mechanism for Counting.). *Let $X$ be a dataset. Then, the mechanism $M(X) = |X| + \mathrm{Lap}(1/\varepsilon)$ is $(\varepsilon, 0)$-DP, where $\mathrm{Lap}(1/\varepsilon)$ is a variable following a Laplace distribution with variance $1/\varepsilon$.*

We use the Gaussian mechanism for more general purposes (e.g., the PCA step). It is defined as follows:

**Lemma 6.** *Gaussian Mechanism Let $f : \mathcal{X} \to \mathbb{R}^n$ be a function with $\ell_2$-sensitivity $\Delta_{f,2}$. Then, for $\sigma(\varepsilon, \delta) = \frac{\sqrt{2 \log(2/\delta)}}{\varepsilon}$ the Gaussian mechanism $M(X) = f(X) + \mathcal{N}_d \left( 0, \Delta_{f,2}^2 \sigma(\varepsilon, \delta)^2 \right)$ is $(\varepsilon, \delta)$-DP, where $\mathcal{N}_d(0, \sigma^2)$ is a d-dimensional Gaussian random variable, where each dimension is independent with mean 0 and variance $\sigma^2$.*

Combining those two mechanisms gives a private and accurate estimate for the average of a dataset

**Lemma 7** (Private averaging). *For dataset $X$ in the ball $B(0, \Delta)$, the mechanism $M(X) := \frac{\sum_{x \in X} X + \mathcal{N}_d(0, \Delta_{f,2} \sigma^2(\varepsilon/2, \delta))}{|X| + \mathrm{Lap}(2/\varepsilon)}$ is $(\varepsilon, \delta)$-DP. Additionally, $|X| \geq$, then it holds with probability $1 - \beta$ that $\|M(X) - \mu(X)\|_2 \leq \frac{\Delta \ln(2/\beta)}{|X|\varepsilon} + \frac{\Delta \sigma(\varepsilon/2, \delta) \sqrt{\ln(2/\beta)}}{|X|}$.*

### B.2. Differentially Private tools for Gaussian mixtures

First, we review some properties of the private rank-$k$ approximation: this algorithm was analyzed by Dwork et al. (2014), and its properties when applied on Gaussian mixtures by Kamath et al. (2019). The guarantee that is verified by the projection onto the noisy eigenvectors is the following:

**Definition 8.** *Fix a matrix $X \in \mathbb{R}^{d \times n}$, and let $\Pi_k$ be the projection matrix onto the principal rank-k subspace of $XX^T$. For some $B \geq 0$, we say that a matrix $\Pi$ is a B-almost k-PCA of $X$ if $\Pi$ is a projection such that:*

- $\|XX^T - (\Pi X)(\Pi X)^T\|_2 \leq \|XX^T - (\Pi_k X)(\Pi_k X)^T\|_2 + B$, *and*
- $\|XX^T - (\Pi X)(\Pi X)^T\|_F \leq \|XX^T - (\Pi_k X)(\Pi_k X)^T\|_F + kB$.

Dwork et al. (2014) shows how to compute a $B$-almost $k$-PCA, with a guarantee on $B$ that depends on the diameter of the dataset:

**Theorem 9** (Theorem 9 of Dwork et al. (2014)). *Let $X \in \mathbb{R}^{d \times n}$ such that $\|X_i\|_2 \leq 1$, and fix $\sigma(\varepsilon, \delta) = \sqrt{2 \ln(2/\delta)}/\varepsilon$. Let $E \in R^{d \times d}$ be a symmetric matrix, where each entry $E_{i,j}$ with $j \geq i$ is an independent draw from $\mathcal{N}(0, \sigma(\varepsilon, \delta)^2)$. Let $\Pi_k$ be the rank-k approximation of $XX^T + E$.*

*Then, $\Pi_k$ is a $O(\sqrt{d} \cdot \sigma(\varepsilon, \delta))$-almost k-PCA of $X$, and is $(\varepsilon, \delta)$-DP.*

Kamath et al. (2019) shows crucial properties of Gaussian mixtures: first, the projection of each empirical mean with a $B$-almost $k$-PCA is close to the empirical mean:

**Lemma 10** (Lemma 3.1 in Kamath et al. (2019)). *Let $X \in \mathbb{R}^{d \times n}$ be a collection of points from k clusters centered at $\mu_1, ..., \mu_k$. Let $C$ be the cluster matrix, namely $C_j = \mu_i$ if $X_j$ belongs to the i-th cluster, and $G_i$ be the i-th cluster.*

*Let $\Pi_k$ be a B-almost k-PCA, and denote $\bar{\mu}_1, ..., \bar{\mu}_k$ the empirical means of each cluster, and $\tilde{\mu}_1, ..., \tilde{\mu}_k$ the projected empirical means.*

*Then, $\|\bar{\mu}_i - \tilde{\mu}_i\| \leq \frac{1}{\sqrt{|G_i|}} \|X - C\|_2 + \sqrt{\frac{B}{|G_i|}}$.*

Second – and this helps bounding the above – they provide bounds on the spectral norm of the clustering matrix $X - C$:

**Lemma 11** (Lemma 3.2 in Kamath et al. (2019))**.** *Let $X \in \mathbb{R}^{d \times n}$ be a set of $n$ samples from a mixture of $k$ Gaussians. Let $\sigma_i$ be the maximal unidirectional variance of the $i$-th Gaussian, and $\sigma_{max} = \max \sigma_i$. Let $C$ be the cluster matrix, namely $C_j = \mu_i$ if $X_j$ is sampled from $\mathcal{N}(\mu_i, \Sigma_i)$.*

*If $n \geq \frac{1}{w_{min}} (\zeta_1 d + \zeta_2 \log_2(k/\beta))$, where $\zeta_1, \zeta_2$ are some universal constants, then with probability $1 - \beta$ it holds that*

$$\frac{\sqrt{n w_{min}} \sigma_{max}}{4} \leq \|X - C\|_2 \leq 4 \sqrt{n \sum_{i=1}^{k} w_i \sigma_i^2}.$$

### B.3. Properties of Gaussian mixtures

**Lemma 12.** *Consider a set $P$ of $n$ samples from a Gaussian mixtures $\{(\mu_i, \Sigma_i, w_i)\}_{i \in [k]}$. Let $G_i$ be the set of points sampled from the $i$-th component. If $n \geq \frac{24 \log(k)}{w_{min}}$, then with probability $0.99$ it holds that $\forall i, |G_i| \geq n w_i / 2$*

*Proof.* This is a direct application of Chernoff bounds: each sample $s$ is in $G_i$ with probability $w_i$. Therefore, the expected size of $G_i$ is $n w_i$, and with probability at least $1 - 2 \exp(-n w_i / 12)$ it holds that $\left| |G_i| - n w_i \right| \leq n w_i / 2$: for $n \geq 24 \log(k) / w_{\min}$, the probability is at least $1 - 2/k^2$. A union-bound over all $i$ concludes. $\qquad \square$

### B.4. Clustering preliminaries

Our algorithm first replaces the full dataset $P$ with a weighted version of $Q$, and then computes a $k$-means solution on this dataset. The next lemma shows that, if $\mathrm{cost}(P, Q)$ is small, then the $k$-means solution on the weighted $Q$ is a good solution for $P$:

**Lemma 13.** *Let $P, C_1 \subset \mathbb{R}^d$, and $f : P \to C_1$ be a mapping with $\Gamma := \sum_{p \in P} \|p - f(p)\|^2$. Let $w_\nu$ be such that $|w_\nu - |f^{-1}(\nu)|| \leq |f^{-1}(\nu)|/2$. Let $\tilde{P}$ be the multiset where each $\nu \in C_1$ appears $w_\nu$ many times, . Let $C_2$ be such that $\mathrm{cost}(\tilde{P}, C_2) \leq \alpha OPT(\tilde{P})$. Then,*

$$\mathrm{cost}(P, C_2) \leq (2 + 12\alpha)\Gamma + 12\alpha OPT(P).$$

*Proof.* Recall that $C_2(p)$ is the closest point in $C_2$ to $p$. We have, using triangle inequality:

$$
\begin{aligned}
\mathrm{cost}(P, C_2) &= \sum_{p \in P} \|p - C_2(p)\|^2 \\
&\leq \sum_{p \in P} \|p - C_2(f(p))\|^2 \\
&\leq \sum_{p \in P} (\|p - f(p)\| + \|f(p) - C_2(f(p))\|)^2 \\
&\leq \sum_{p \in P} 2\|p - f(p)\|^2 + 2\|f(p) - C_2(f(p))\|^2 \\
&\leq 2\Gamma + 2 \sum_{\nu \in C_1} |f^{-1}(\nu)| \|\nu - C_2(\nu)\|^2 \\
&\leq 2\Gamma + 4 \sum_{\nu \in C_1} w_\nu \|\nu - C_2(\nu)\|^2 \\
&\leq 2\Gamma + 4\alpha OPT(\tilde{P}).
\end{aligned}
$$

A similar argument bounds $OPT(\tilde{P})$: let $C^*$ be the optimal solution for $P$, then, for any point $p$ we have $\|f(p) - C^*(f(p))\| \leq$

$\|f(p) - C^*(p)\| \le \|f(p) - p\| + \|p - C^*(p)\|$. Therefore,

$$
\begin{aligned}
\mathrm{OPT}(\tilde{P}) &\le \sum_{\nu \in C_1} w_\nu \|\nu - C * (\nu)\|^2 \\
&\le \frac{3}{2} \sum_{\nu \in C_1} |f^{-1}(\nu)| \|\nu - C * (\nu)\|^2 \\
&\le 3 \sum_{p \in P} 2\|C_1(p) - p\|^2 + 2\|p - C^*(p)\|^2 \\
&\le 3\Gamma + 3\mathrm{OPT}(P).
\end{aligned}
$$

Combining those two inequalities concludes the lemma. $\qquad\square$

## C. The non-private, non-federated algorithm of Awasthi & Sheffet (2012)

The algorithm we take inspiration from is the following, from Awasthi & Sheffet (2012) and inspired by Kumar & Kannan (2010): first, project the dataset onto the top-$k$ eigenvectors of the dataset, and compute a constant-factor approximation to $k$-means (e.g., using local search). Then, improve iteratively the solution with Lloyd's steps. The pseudo-code of this algorithm is given in Algorithm 3, and the main result of Awasthi & Sheffet (2012) is the following theorem:

**Theorem 14** (Awasthi & Sheffet (2012))**.** *For a separated Gaussian mixture, Algorithm 3 correctly classifies all point w.h.p.*

Their result is more general, as they do not require the input to be randomly generated, and only requires a strict separation between the clusters. In this paper, we focus specifically on Gaussian mixtures.

---
**Algorithm 3** Cluster($P$)
---
1: **Part 1:** find initial Clusters

    a) Compute $\hat{P}$ the projection of $P$ onto the subspace spanned by the top $k$ singular vectors of $P$.

    b) Run a $c$-approximation algorithm for the $k$-means problem on $\hat{P}$ to obtain centers $\nu_1, ..., \nu_k$.

2: **Part 2:** For $r = 1, ...k$, set $S_r \leftarrow \{i : \forall s, \|\hat{P}_i - \nu_r\| \le \frac{1}{3}\|\hat{P}_i - \nu_s\|\}$ and $\theta_r \leftarrow \mu(S_r)$

3: **Part 3:** Repeat Lloyd's steps until convergence:
    for $r = 1, ...k$, set $C(\nu_r) \leftarrow \{i : \forall s, \|P_i - \nu_r\| < \|P_i - \nu_s\|\}$, and $\theta_r \leftarrow \mu(C(\nu_r))$

---

## D. Our result

Our main theoretical results is to adapt Algorithm 3 to a private and federated setting. As mention in the main body, we show a stronger version of Theorem 3, without assumptions on diameter and server data. More precisely, we show the following theorem:

**Theorem 15.** *There is an $(\varepsilon, \delta)$-DP algorithm with the following accuracy guarantee. Suppose that the client dataset $P$ is generated from a separated Gaussian mixtures with $n \ge \zeta_1 \frac{kdT \log^2 n \cdot \sqrt{\ln(1/\delta)}}{\varepsilon^2 w_{min}^2}$ samples, where $\zeta_1$ is some universal constant, and that $Q$ contains a least one sample from component of the mixture.*

*Then, the algorithm computes centers $\nu_1, ..., \nu_k$ such that, for some universal constants $\zeta_2, \zeta_3$, after $T + \zeta_2 \log \frac{\sigma_{max} \log |Q|}{\varepsilon w_{min}}$ rounds of communications, it holds with high probability that:*

$$
\|\mu_i - \nu_i\| \le \zeta_3 \max\left( \frac{1}{2^T}, \frac{kdT \log^2 n \sigma_{max} \sqrt{\ln(T/\delta)}}{n\varepsilon^2 w_{min}^2} \right).
$$

Note that the precision increases with the number of samples: if $n$ is larger than $\frac{2^T \log(\sigma_{\max}/w_{\min}) kd \log^2 n \sigma_{max}}{\varepsilon^2 w_{\min}^2}$, then the dominating term is $1/2^T$.

**Corollary 16.** *There is an $(\varepsilon, \delta)$-DP algorithm with $O(\log(n))$ rounds of communications with the following accuracy guarantee. Suppose that the client dataset $P$ is generated from a separated Gaussian mixtures with $n = \Omega\left(\frac{k \log^2 n \sqrt{d} \sigma_{max}}{\varepsilon^2 w_{min}^2}\right)$ samples, that $Q$ contains a least one sample from each component of the mixture and at most $n$ data points. Suppose that $n = \Omega\left(\frac{k \log^3 n \sqrt{d} \sigma_{max}}{\varepsilon^2 w_{min}^2}\right)$, and that $n = \Omega\left(\frac{\log(n)^6 \cdot k d^2}{\varepsilon^4 w_{min}^2}\right)$.*

*Then, the algorithm computes centers $\nu_1, ..., \nu_k$ such that, with high probability, the clustering induced by $\nu_1, ..., \nu_k$ is the partition $G_1, ..., G_k$.*

*Proof.* Theorem 5.4 of Kumar & Kannan (2010) (applied to Gaussian mixtures) bounds the number of misclassified points in a given cluster in terms of the distance between $\nu_i$ and $\mu_i$. Define, for any $i$, $S_i$ as the cluster of $\nu_i$, and $\delta_i = \|\mu_i - \nu_i\|$. Then, for $j \neq i$, Kumar & Kannan (2010) show that, for some constant $c'$:

$$|G_i \cap S_j| \leq \frac{c' n w_{min}(\delta_i^2 + \delta_j^2)}{\|\mu_i - \mu_j\|^2}{}_4$$

Since $\|\mu_i - \mu_j\|^2 \geq c^2 \frac{k \sigma_{max}^2 \log(n)^2}{w_{min}}$, we get that the number of points from $G_i$ assigned to cluster $j$ is at most $\frac{c' n w_{min}^2(\delta_i^2 + \delta_j^2)}{k \sigma_{max}^2 \log(n)^2}$.

We aim at bounding $\delta_i$ and $\delta_j$ using Theorem 15. For $T = \log\left(\frac{10 c' n w_{min}}{k \sigma_{max}}\right)$, it holds that $\frac{1}{2^T} \leq \frac{\sqrt{k} \sigma_{max}}{10 c' \sqrt{n} w_{min}}$.

In addition, for this value of $T$ and a number of samples $n$ at least $n \geq \frac{100 c'^2 \log(n)^2 \cdot k d^2 \log(n)^4}{\varepsilon^4 w_{min}^2}$, we also have $\frac{k d T \log^2 n \sigma_{max} \sqrt{\ln(T/\delta)}}{n \varepsilon^2 w_{min}^2} \leq \frac{\sqrt{k} \sigma_{max}}{10 c' \sqrt{n} w_{min}}$.

Therefore, the upper bound on $\delta_i$ and $\delta_j$ from Theorem 15 after $T + \log\left(\sigma_{max} \log|Q|/w_{min}\right) = O(\log(n))$ rounds of communications ensure that there is no point misclassified. This which concludes the statement. $\square$

In the case where the assumption of Theorem 3 are satisfied, namely, (1) the diameter is bounded and (2) the server data are well spread, then the algorithm of Theorem 15 reduces directly to Algorithm 1 followed with $T$ steps of Algorithm 2, with only $T$ rounds of communication. Indeed, the first $O\left(\log \frac{\sigma_{max} \log|Q|}{\varepsilon w_{min}}\right)$ rounds of the algorithm from Theorem 15 are dedicated to enforcing condition (1) and (2): if they are given, there is no need for those steps.

The organization of the proof is as follows. First, we give some standard technical preliminary tools about differential privacy and Gaussian mixtures. Then, we show how to implement Algorithm 3: the bulk of the work is in the implementation of its Part 1, computing a good solution for $\Pi P$. The second part to iteratively improve the solution is very similar to the non-private part.

# E. Part 1: Computing centers close to the means

### E.1. Reducing the diameter

**Lemma 17.** *There is an $\varepsilon$-DP algorithm with one communication round that, given $w_{min}$ and $\sigma_{max}$, reduces the diameter of the input to $O\left(\frac{\log|Q| \log n \sqrt{d} \sigma_{max}}{\varepsilon w_{min}}\right)$.*

*Proof.* We fix a distance $D = 4 \log n \sqrt{d} \sigma_{max}$. First, the server identifies regions that contains many server points: if $q$ is such that $|Q \cap B(q, D)| \geq \frac{\varepsilon n w_{min}}{200 \log|Q|}$, then $q$ is marked *frozen*.

Then, each client assigns its points to their closest server point in $Q$, breaking ties arbitrarily. In one round of communication, the server learns, for each server point $q \in Q$, the noisy number of points assigned to $q$, namely $\hat{w}_q(P) = w_q(P) + \text{Lap}(1/\varepsilon)$. For privacy, the noise added to each count follows a Laplace distribution with parameter $1/\varepsilon$. Hence, with high probability, the noise is at most $O\left(\frac{\log|Q|}{\varepsilon}\right)$ on each server data $q$.

---

[4]We simplified the original statement of Kumar & Kannan (2010) to directly adapt it to separated Gaussian mixtures: in this case, $\|P - C\|_2^2 \leq 4 n \sigma_{max}^2$, and $\Delta_{i,j}$ (defined in the original statement) is our separation value, $c \sqrt{k/w_{min}} \sigma_{max} \log(n)$.

With high probability on the samples, for all $i$ the $B(\mu_i, D)$ contains all the $w_i n$ samples from $\mathcal{G}_i$. Therefore, any server point $q$ sampled from $\mathcal{G}_i$ is either frozen, or the noisy count in $B(q, D)$ ball is at least $nw_{\min}/2 - |Q \cap B(\mu_i, D)| \cdot \frac{\log |Q|}{\varepsilon} \geq nw_{\min}/3$, using Lemma 12.

Consider now an arbitrary point $p \in \mathbb{R}^d$. Since $G_i$ is fully contained in $B(\mu_i, D/2)$, either the ball $B(p, D/2)$ doesn't intersect with $G_i$, or $B(p, D)$ contains entirely $G_i$. Furthermore, by triangle inequality, for any $q \in G_i \cap Q$ the ball $B(q, D)$ contains entirely $G_i$: if $q$ is not frozen, it has noisy count at least $nw_{\min}/3$, and therefore true count at least $nw_{\min}/6$.

To reduce the diameter, we first remove all points from $Q$ that are not frozen and for which the ball $B(q, D)$ has noisy count less than $nw_{\min}/3$: by the previous discussion, those points are not sampled from any $\mathcal{G}_i$ and are part of the noise. In addition, connect any pair of points that are at distance less than $D$.

We claim that each connected component has diameter at most $O\left(\frac{\log^2 n \sqrt{d} \sigma_{\max}}{\varepsilon w_{\min}}\right)$.

To prove this claim, we fix such a component, and consider the following iterative procedure. Pick an arbitrary point from the component, and remove all points that are at distance $2D$. Repeat those two steps until there are no more points.

Let $q$ be a point selected at some step of this procedure. First, note that $B(q, D)$ is disjoint from any ball $B(q', D)$, for $q'$ previously selected – as $B(q', 2D)$ has been removed. Furthermore, either $q$ is frozen and the ball contains $\frac{\varepsilon n w_{\min}}{200 \log |Q|}$ many points of $Q$, or $q$ is not frozen and $B(q, D)$ contains at least $nw_{\min}/6$ points of $P$. Therefore, there are at most $t_{max} := \frac{6}{w_{\min}} + \frac{200 \log |Q|}{\varepsilon w_{\min}}$ iterations. So the connected component can be covered with $t_{max}$ balls of radius $2D$. Additionally, since each edge has length at most $D$, the component has diameter at most $O(t_{max} D) = O\left(\frac{\log |Q| \log n \sqrt{d} \sigma_{\max}}{\varepsilon w_{\min}}\right)$. This concludes the claim.

The other key property of the connected component is that each $G_i$ is fully contained in a single connected component, as all points of $G_i$ are at distance at most $D$ of each other.

Therefore, we can transform the space such that the connected components get closer but do not interact, so that the diameter reduces while the centers of Gaussians are still far apart. Formally, let $D'$ be the maximum diameter of the connected components. Select an arbitrary representative in $Q$ from each connected component, and apply a translation to the connected component such that its representative has coordinate $(100D' \cdot i, 0, 0, ..., 0)$. This affine transformation ensures that (1) within each connected component, all means are still separated and the points are still drawn from Gaussian with the same covariance matrix and (2) the separation between centers of different component is at least $50D'$.

Therefore, the instance constructed still satisfy the separation conditions of Definition 2, and has diameter at most $O(kD') = O\left(\frac{k \log n \log |Q| \sqrt{d} \sigma_{\max}}{\varepsilon w_{\min}}\right)$ $\qquad \square$

### E.2. A relaxation of Awathi-Sheffet's conditions

The result of Awasthi & Sheffet (2012), applied to Gaussian, requires a slightly weaker separation between the centers than what we enforce. They consider a dataset $P$ sampled from a Gaussian mixtures, and with cluster matrix $C$ (namely, $C_i = \mu_i$ if $P_i$ is sampled from the $i$-th component). They define for each cluster $\Delta_i^{AS} := \frac{1}{\sqrt{|G_i|}} \min(\sqrt{k} \|P - C\|_2), \|P - C\|_F)$, and require $\|\mu_i - \mu_j\| \geq c(\Delta_i^{AS} + \Delta_j^{AS})$ for some large constant $c$.

In the Gaussian setting, we have $|G_i| \approx nw_i$ (Lemma 12), $\|P - C\|_2 = O(\sigma_{\max} \sqrt{n})$ Lemma 11 and $\|A - C\|_F = \Theta(\sqrt{nd} \sigma_{\max})$. Thus, in most cases, $\min\left(\sqrt{k} \|P - C\|_2, \|P - C\|_F\right) = \sqrt{nk} \sigma_{\max} \operatorname{polylog}(d/w_{\min})$, except in some degenerate cases – and we keep the minimum only to fit with the proof of Awasthi & Sheffet (2012).

We can define $\Delta_i = \frac{\sigma_{\max} \sqrt{n}}{\sqrt{|G_i|}} \min\left(\sqrt{k} \operatorname{polylog}(d/w_{\min}), \sqrt{d}\right)$: our separation condition Definition 2 ensures that $\|\mu_i - \mu_j\| \geq c(\Delta_i + \Delta_j)$, for some large $c$. We now show the two key lemmas from Awasthi & Sheffet (2012), adapted to our private algorithm.

**Fact 18** (Fact 1.1 in Awasthi & Sheffet (2012)). *Let $P \in \mathbb{R}^{d \times n}$ be a set of $n$ points sampled from a Gaussian mixtures, and let $C$ be the cluster matrix, namely the $j$-th column is $C_j = \mu_i$ if $X_j$ is sampled from $\mathcal{N}(\mu_i, \Sigma_i)$. Let $\Pi$ be a $B$-almost $k$-PCA for $P_1, ..., P_n$. Suppose that $B$ satisfies $B \leq \frac{\sqrt{nw_{min}} \sigma_{max}}{4k}$. Then:*

$$\|\Pi P - C\|_F^2 \leq 20 \min(k \|A - C\|_2^2, \|A - C\|_F^2)(= nw_i \Delta_i^2).$$

*Proof.* First, since both $\Pi P$ and $C$ have rank $k$, it holds that $\|\Pi P - C\|_F^2 \leq 2k\|\Pi P - C\|_2^2$. By triangle inequality, this is at most $2k\left(\|\Pi P - P\|_2 + \|P - C\|_2\right)^2$.

Now, we have that $\|\Pi P - P\|_2^2 = \|(\Pi P - P)^T(\Pi P - P)\|_2$: since $\Pi$ is an orthogonal projection, $\Pi = \Pi^T = \Pi^2$ and therefore $\|\Pi P - P\|_2^2 = \|PP^T - (\Pi P)(\Pi P)^T\|_2$. Using that $\Pi$ is a $B$-almost $k$-PCA of $P$, this is at most $\|(\Pi_k P - P)(\Pi_k P - P)^T\|_2 + B$, where $\Pi_k P$ is the best rank-$k$ approximation to $P$. By definition of $\Pi_k$, this is equal to $\|P - \Pi_k P\|_2^2 + B \leq \|P - C\|_2^2 + B$.

Overall, we get using $\sqrt{a + b} \leq \sqrt{a} + \sqrt{b}$:

$$\|\Pi P - C\|_F^2 \leq 2k\left(\|\Pi P - P\|_2 + \|P - C\|_2\right)^2$$
$$\leq 2k\left(2\|P - C\|_2 + \sqrt{B}\right)^2$$
$$\leq 16k\|P - C\|_2^2 + 4kB.$$

Using Lemma 11 and the assumption that $4kB \leq \sqrt{nw_{\min}}\sigma_{\max}$ concludes the first part of the lemma.

For the other term, we have $\|\Pi P - C\|_F \leq \|\Pi P - P\|_F + \|P - C\|_F$. The fact that $\Pi$ is a $B$-almost $k$-PCA ensures that $\|\Pi P - P\|_F^2 \leq \|P - C\|_F^2 + kB$; and the fact that $\|P - C\|_F^2 \geq \|P - C\|_2^2 \geq \frac{nw_{\min}\sigma_{\max}^2}{16} \geq B$ concludes (where the second inequality is from Lemma 11). $\qquad\square$

**Fact 19.** *[Analogous to Fact 1.2 in* Awasthi & Sheffet (2012)*] Let $P \in \mathbb{R}^{d \times n}$ be a Gaussian mixtures, and $\Pi$ be a $B$-almost $k$-PCA for $P_1, ..., P_n$. Suppose that $B$ satisfies $B^2 \leq nw_{min}\sigma_{max}^2$. Let $S = \{\nu_1, ..., \nu_k\}$ be centers such that $\mathrm{cost}(\Pi P, S) \leq nk\sigma_{max}^2 \cdot \log^2 n$.*

*Then, for each $\mu_i$, there exists $j$ such that $\|\mu_i - \nu_j\| \leq 6\Delta_i$, so that we can match each $\mu_i$ to a unique $\nu_j$.*

*Proof.* The proof closely follows the one in Awasthi & Sheffet (2012). Assume by contradiction that there is a $i$ such that $\forall j, \|\mu_i - \nu_j\| > 6\Delta_i$. For any point $p \in P$, let $\nu_p$ be its closest center. Then, the contribution of the points in $G_i$ to the cost is at least

$$\sum_{p \in G_i} \|\mu_i - \nu_p + \Pi p - \mu_i\|^2 > \frac{|G_i|}{2}(6\Delta_i)^2 - \sum_{p \in G_i} \|\Pi p - \mu_i\|^2 \geq 18|G_i|\Delta_i^2 - \|\Pi P - C\|_F^2,$$

where the first inequality follows from $(a - b)^2 \geq \frac{a^2}{2} - b^2$. Using first that $|G_i|\Delta_i^2 = 100nk\sigma_{\max}^2 \log^2(n)$, then Fact 18 combined with Lemma 11 yields that $\sum_{p \in G_i} \|\Pi p - \nu_p\|^2 > 1800nk\sigma_{\max}^2 \log^2(n) - 16nk\sigma_{\max}^2$ This contradicts the assumption on the clustering cost. $\qquad\square$

Assuming there is a matching as in Fact 19, the proof of Awasthi & Sheffet (2012) directly goes through (when the Lloyd steps in Parts 2 and 3 of the algorithm are implemented non-privately), and we can conclude in that case that the clustering computed by Algorithm 1 is correct. Therefore, we first show that our algorithm computes a set of centers satisfying the conditions of Fact 19; and will show afterwards that the remaining of the proof works even with the addition of private noise.

### E.3. Computing a good $k$-means solution for $\Pi P$

The goal of this section is to show the following lemma:

**Lemma 20.** *There is an $\varepsilon$-DP algorithm with $10\log\frac{4\log|Q|}{\varepsilon w_{min}}$ rounds of communications that computes a $k$-means solution $S$ with*

$$\mathrm{cost}(\Pi P, S) = O\left(n \cdot \log^2\left(\frac{1}{\varepsilon w_{min}}\right) \cdot k\sigma_{max}^2 \log n\right).$$

The proof of this lemma is divided into several parts: first, we show that the means of the projected Gaussians $\Pi\mu_1, ..., \Pi\mu_k$ would be a satisfactory clustering. As points in $Q$ are drawn independently from $\Pi$, there are points $\Pi Q$ close to each center $\Pi\mu_i$: our second step is therefore an algorithm that finds those points, in few communications rounds.

**Lemma 21.** *Let $\Pi$ be the private projection computed by the algorithm. With high probability, clustering the projected set $\Pi G_i$ to the projected mean $\Pi\mu_i$ has cost $|G_i|\log n \cdot k\sigma_{max}^2$.*

*Proof.* We focus on a single Gaussian $\mathcal{G}_i$, and denote for simplicity $\mu := \mu_i$ its center and $\hat{\Sigma} := \Pi\Sigma_i$ the covariance matrix of $\Pi\mathcal{G}_i$. Standard arguments (see Proof of Corollary 5.15 in Kamath et al. (2019), or the blog post from McSherry (2014)) show that, with high probability, for all point it holds that $\|\Pi(p - \mu)\|_2^2 \leq \sqrt{k\log(n)}\sigma_{\max}$.

For a sketch of that argument, notice that if the projection $\Pi$ was fixed independently of the samples, this inequality is direct from the concentration of Gaussians around their means, as the projection of $\mathcal{G}_i$ via $\Pi$ is still a Gaussian, with maximal unidirectional variance at most $\sigma_{\max}$. This does not stay true when $\Pi$ depends on the sample; however, since $\Pi$ is computed via a private mechanism, the dependency between $\Pi$ and any fixed sample is limited, and we can show the concentration.

Combined with the fact that there are $|G_i|$ samples from $\mathcal{G}_i$, this concludes. □

Lemma 20 in particular ensures that clustering $\Pi P$ to the full set $\Pi Q$ yields a cost $nk\sigma_{\max}^2 \cdot \log^2 n$. Therefore, if we could compute for each $q \in Q$ the size $w_q(\Pi P)$ of $\Pi q$'s cluster in $\Pi P$, namely, the number of points in $\Pi P$ closer to $\Pi q$ than to any other point in $\Pi Q$ (breaking ties arbitrarily), then Lemma 13 would ensure that computing an $O(1)$-approximation to $k$-means on this weighted set yields a solution to $k$-means on $\Pi P$ with cost $O\left(nk\sigma_{\max}^2 \cdot \log^2 n\right)$.

However, the privacy constraint forbids to compute $w_q(\Pi P)$ exactly, and the server only receives a noisy version $\widehat{w_q(\Pi P)}$ – with a noise following a Laplace noise with parameter $1/\varepsilon$. Hence, for all points $q \in Q$, the noise added is at most $\frac{\log n}{\varepsilon}$ with high probability.

### E.4. If assumption (2) is satisfied: the noise is negligible

Assumption (2) can be used to bound the total amount of noise added to the server data: we can show that the total contribution of the noise is small compared to the actual $k$-means cost, in which case solving $k$-means on the noisy data set yields a valid solution. We show the next lemma:

**Lemma 22.** *For any set of $k$ centers $S$, it holds that* $\left|\sum_q w_q(\Pi P)\cos(p, S) - \sum_q \widehat{w_q(\Pi P)}\cos(p, S)\right| \leq \frac{|Q|\log|Q|\Delta^2}{\varepsilon}$

*Proof.*

$$\left|\sum_q w_q(\Pi P)\cos(q, S) - \sum_q \widehat{w_q(\Pi P)}\cos(q, S)\right| = \left|\sum_q \mathrm{Lap}(1/\varepsilon)\cos(q, S)\right|$$

With high probability, each of the $|Q|$ Laplace law is smaller than $\frac{\log|Q|}{\varepsilon}$. In this case, we get $\left|\sum_q \mathrm{Lap}(1/\varepsilon)\cos(q, S)\right| \leq$ $\frac{|Q|\log|Q|\Delta^2}{\varepsilon}$ Therefore, the gap between the solution evaluated with true weight $w_q(\Pi P)$ and noisy weight $\widehat{w_q(\Pi P)}$ is at most $\frac{|Q|\log|Q|\Delta^2}{\varepsilon}$ □

Using $|Q| \leq n$, the assumption $|Q| \leq \frac{\varepsilon nk\sigma_{\max}^2}{\Delta^2}$ therefore ensures that the upper bound of the previous lemma is at most $nk\log(n)\sigma_{\max}^2$.

Hence, if $S$ is a solution that has cost $O(1)$ times optimal on the noisy projected server data, it has cost $O(nk\sigma_{\max}^2\log(n))$ on the projected server data. Combining this result with Lemma 13 concludes: $\cos(\Pi P, S) = O\left(nk\sigma_{\max}^2\log(n)\right)$.

### E.5. Enforcing Assumption (2)

In order to get rid of Assumption (2), we view the problem slightly differently: we will not try to reduce the number of points in $Q$ to the precise upper-bound, but will nonetheless manage to control the noise and show Lemma 20.

Indeed, if all points of $Q$ get assigned more than $2\log n/\varepsilon$ many input points, then the estimates of $w_q$ are correct up to a factor 2, and Lemma 13 shows that a $k$-means solution $S$ for the dataset consisting of $\Pi Q$ with the noisy weights satisfies $\cos(\Pi P, S) = O\left(nk\sigma_{\max}^2 \cdot \log^2 n\right)$. However, it may be that some points of $Q$ get assigned less than $2\log n/\varepsilon$ points, in which case the noise would dominate the signal and Lemma 13 becomes inapplicable. Our first goal is therefore to preprocess the set of hings $Q$ to get $\hat{Q}$ such that :

1. for each cluster, $\Pi\hat{Q}$ still contains one good center, and

2. $\forall q \in \hat{Q}, \hat{w}_q \geq 2 \log n/\varepsilon$ (where the weight $\hat{w}$ is computed by assigning each data point to its closest center of $\hat{Q}$)

The first item ensures that $\mathrm{cost}(\Pi P, \Pi \hat{Q}) = O\left(nk\sigma_{\max}^2 \cdot \log^2 n\right)$; the second one that the size of each cluster is well approximated, even after adding noise.

Our intuition is the following. Removing all points $q \in Q$ with estimated weight less than $2 \log n/\varepsilon$ is too brutal: indeed, it may be that one cluster is so over-represented in $Q$ that all its points get assigned less than $2 \log n/\varepsilon$ points from $P$. However, in that case, there are many points in the cluster and in $\Pi Q$: we can therefore remove each point with probability $1/2$ and preserve (roughly) the property that there is a good center in $\Pi Q$. Repeating this intuition, we obtain the algorithm described in Algorithm 4.

---

**Algorithm 4** SimplifyServerData

1: **Input:** Server data $Q$, client datasets $P^1, ..., P^m$, a projection matrix $\Pi$ computed from $P^1, ..., P^m$, and privacy parameter $\varepsilon$
2: Let $F \leftarrow \emptyset, Q_0 \leftarrow Q, T = 10 \log \left(\frac{4 \log |Q|}{\varepsilon w_{\min}}\right)$
3: **for** $t = 0$ to $T$ **do**
4:      Let $C = F \cup Q_t$
5:      for each $q \in C$, the server receives a noisy estimate $\hat{w}_q^{(t)}$ of $w_{\Pi q}(\Pi Q_t)$, with noise $\mathrm{Lap}(T/\varepsilon)$.
6:      Server computes $L := \{q \in C : \hat{w}_q^{(t)} \leq 2 \log n/\varepsilon\}$.
7:      $F \leftarrow F \cup (Q_t \setminus L)$.
8:      Server computes $Q_{t+1}$, a subset of $L$ where each point is sampled with probability $1/2$.
9: **end for**
10: **Return:** $F$

---

We sketch briefly the properties of algorithm 4, before diving into details of the proof. First, the algorithm is $\varepsilon$-DP, as each of the $T$ steps is $\varepsilon/T$-DP.

Then, points in $F$ are *frozen*: even after adding noise, their weight is well approximated. We will show by induction on the time $t$ that, for any cluster $i$ that does not contain any frozen point at time $t$, then $Q_t \cap B(\mu_i, 2t \cdot \sqrt{k \log n}\sigma_{\max})$ contains many points: more precisely, $|Q_t \cap B(\mu_i, 2t \cdot \sqrt{k \log n}\sigma_{\max})| \geq \varepsilon |G_i|/2$. Since at each time step only half of the points in $L$ are preserved in $Q_{t+1}$ (line 7 of the algorithm), it implies that, at the beginning, $|Q \cap B(\mu_i, 2t \cdot \sqrt{k \log n}\sigma_{\max})| \gtrsim 2^t \varepsilon/T |G_i|$. Therefore, for $t = \log(1/(\varepsilon w_{\min}))$, we have for each cluster that either it contains a frozen point, or $|Q \cap B(\mu_i, 2t \cdot \sqrt{k \log n}\sigma_{\max})| \geq \frac{|G_i|}{w_{\min}} > n$: as the second option is not possible, all clusters contains a frozen point, which is a good center for that cluster.

Our next goal is to formalize the argument above, and show:

**Lemma 23.** *Let $F$ be the output of Algorithm 4. Then, for each cluster $i$, there is a point $\nu_i \in F$ such that $\|\Pi(\mu_i - \nu_i)\| \leq \log \left(\frac{4 \log |Q|}{\varepsilon w_{min}}\right) \cdot \sqrt{k \log n}\sigma_{max}$.*

*Furthermore, for each $q \in F$, define $w_q$ as the number of points closest to $q$ than any other point in $F$: it holds that $w_q \geq 2 \log n/\varepsilon$.*

For simplicity, we define $\Delta_C := \sqrt{k \log n}\sigma_{\max}$. To prove this lemma, we show inductively that after $t$ iterations of the loop in the algorithm, then either $B(\Pi \mu_i, 2t\Delta_C)$ contains a frozen point, or $|B(\Pi \mu_i, (t+1)\Delta_C) \cap \Pi Q_t| \geq \varepsilon |G_i|/2$. Since the number of points in $\Pi Q_t$ is divided by roughly 2 at every time step, the latter condition implies that there was initially at least $\approx 2^t \varepsilon |G_i|$ points in $B(\Pi \mu_i, (t+1)\Delta_C) \cap \Pi Q$. For $t \approx \log(1/(\varepsilon w_{\min}))$, this is bigger than $n$ and we get a contradiction: the ball contains therefore a frozen point.

Our first observation to show this claim is that many points of $P$ are close to $\mu_i$:

**Fact 24.** *With high probability on the samples, it holds that $\left|B(\Pi \mu_i, \sqrt{k \log n}\sigma_{max}) \cap \Pi P_i\right| \geq |G_i|$*

*Proof.* As in the proof of Lemma 21, the fact that $\Pi$ is computed privately ensures that, with high probability, all points $p \in G_i$ satisfy $\|\Pi(p - \mu_i)\| \leq \sqrt{k \log n}\sigma_{\max}$. Thus, $\left|B(\Pi \mu_i, \sqrt{k \log n}\sigma_{\max}) \cap \Pi P_i\right| \geq |G_i|$. $\square$

For the initial time step $t = 0$ we actually provide a weaker statement to initialize the induction, and show that there is at least one point in $B(\Pi\mu_i, \sqrt{k\log n}\sigma_{max}) \cap \Pi Q_t$. This will be enough for the induction step.

**Fact 25** (Initialization of the induction). *With high probability,* $\exists q \in Q, \|\Pi(\mu_i - q)\| \leq \sqrt{k\log n}\sigma_{max}$.

*Proof.* This directly stems from the fact that there is some point $q \in Q$ that is sampled according to $\mathcal{G}_i$, and that $\Pi$ is independent of that point. Therefore, $\Pi q$ follows the Gaussian law $\Pi\mathcal{G}_i$, which is in a $k$ dimensional space and has maximal unidirectional variance $\sigma_{max}$. Concentration of Gaussian random variables conclude. $\square$

To show our induction, the key lemma is the following:

**Lemma 26.** *After $t$ iteration of the loop, either $B\left(\Pi\mu_i, (t+1)\sqrt{k\log n}\sigma_{max}\right)$ contains a frozen point, or $\left|\Pi Q_t \cap B(\Pi\mu_i, 2(t+1)\sqrt{k\log n}\sigma_{max})\right| \geq \frac{\varepsilon|G_i|}{4T\log(T|Q|)}$.*

*Proof.* Let $\Delta_C := \sqrt{k\log n}\sigma_{max}$.

First, it holds with high probability that all the noise added Line 5 satisfy $\left|\hat{w}_q^{(t)} - w_{\Pi q}(\Pi Q_t)\right| \leq T\log(T|Q|)/\varepsilon$. This directly stems from concentration of Laplace random variables, and the fact that there are $T|Q|$ many of them.

We prove the claim by induction. Fix a $t \geq 0$. The induction statement at time $t$ ensures that either there is a point frozen in $B(\Pi\mu_i, (t+1)\Delta_C)$, in which case we are done, or there is at least one point in $\Pi Q_t \cap B(\Pi\mu_i, (t+1)\Delta_C)$ (note that this statement holds for $t = 0$ by Fact 25).

By triangle inequality, this means that all points of $\Pi G_i \cap B(\Pi\mu_i, \Delta_C)$ are assigned at time $t+1$ to a point in $B(\Pi\mu_i, (t+2)\Delta_C)$ (in line 4 of Algorithm 4). Therefore, by Fact 24, $\sum_{q:\Pi q \in \Pi Q_t \cap B(\Pi\mu_i, (t+2)\Delta_C)} w_q^t \geq |G_i|$.

Then, either $\Pi Q_t$ contains less than $\frac{\varepsilon|G_i|}{2T\log(T|Q|)}$ many points from $B\left(\Pi\mu_i, (t+2)\Delta_C\right)$, and we are done, as one of them must have $w_{\Pi q}(\Pi Q_{t+1}) \geq 2T\log(|Q|T)/\varepsilon$ and will be frozen – as in this case $\hat{w}_q^{(t)} \geq T\log(|Q|T)/\varepsilon$. Or, there are more than $\frac{\varepsilon|G_i|}{2T\log(T|Q|)}$ points, and they all have $w_q^{t+1} \leq 2T\log(T|Q|)/\varepsilon$ : Chernoff bounds ensure that, with high probability, at least $\frac{\varepsilon|G_i|}{4T\log(T|Q|)}$ will be sampled in the set $Q_{t+1}$, which concludes the lemma. $\square$

Lemma 23 is a mere corollary of those results:

*Proof of Lemma 23.* Again, we define $\Delta_C := \sqrt{k\log n}\sigma_{max}$. At the end of Lemma 23, all points in $F$ are frozen: let $f : P \to F$ such that $f(p) = \arg\min_{q \in F} \|\Pi(p - q)\|$, breaking ties arbitrarily. Since all points are frozen, it holds that for all $q, |f^{-1}(q)| \geq 2\log n/\varepsilon$: therefore, their noisy weight $\hat{w}_q$ satisfy $|\hat{w}_q - |f^{-1}(q)|| \leq \frac{|f^{-1}(q)|}{2}$.

Furthermore, for $T$ large enough it holds that $T \geq \log\left(\frac{4T\log(T|Q|)}{\varepsilon w_{min}}\right)$: this holds e.g. for $T = 10\log\left(\frac{4\log(|Q|)}{\varepsilon w_{min}}\right)$.

Lemma 26 ensures that either $B(\Pi\mu_i, (T+1)\Delta_C)$ contains a frozen point, or $|\Pi Q_T \cap B(\Pi\mu_i, 2(T+1)\Delta_C)| \geq \frac{\varepsilon|G_i|}{4T\log(T|Q|)}$.

Suppose by contradiction that we are in the latter case. Since, at each time step, every point in $Q$ is preserved with probability $1/2$, it holds with high probability that $|\Pi Q \cap B(\Pi\mu_i, 2(T+1)\Delta_C)| \geq \varepsilon 2^T \cdot |G_i|$. Indeed, all points of that ball are still present in $Q_T$ with probability $1/T^t$: Chernoff bounds ensure that there must be initially at least $2^T \cdot \frac{\varepsilon|G_i|}{4T\log(T|Q|)}$ points in that ball in order to preserve $\frac{\varepsilon|G_i|}{4T\log(T|Q|)}$ of them after the sampling. With our choice of $T$, this means $|\Pi Q \cap B(\Pi\mu_i, 2(t+1)\Delta_C)| > |Q|$, which is impossible.

Therefore, it must be that $B(\Pi\mu_i, (T+1)\Delta_C)$ contains a frozen point, which concludes the proof. $\square$

**Proof of Lemma 20** We now have all the ingredients necessary to the proof of Lemma 20. The algorithm is a mere combination of the previous results:

- Use Algorithm 4 to compute a set $F$.

- Server sends $F$ to the clients, who define $f : P \to F$ such that $f(p) = \arg\min_{q \in F} \|\Pi(p - q)\|$, breaking ties arbitrarily.

- Client $i$ sends $w_{\Pi q}(\Pi P^i) := \big|\{p \in P^i : f(p) = q\}\big|$.

- Server receives $\hat{w}_q$, a noisy version of $w_q := \sum_i w_q^i$.

- Server computes an $O(1)$-approximation $S$ to $k$-means on the dataset $\Pi F$ with weights $\hat{w}_q$.

To show that $S$ has the desired clustering cost, we aim at applying Lemma 13. For this, we first bound $\sum_p \|\Pi(p - f(p))\|^2$. For each cluster $i$, let $\nu_i$ be the point from $F$ as defined in Lemma 23. We have, using the definition of $f$ and triangle inequality:

$$\sum_p \|\Pi(p - f(p))\|^2 \leq \sum_i \sum_{p \in G_i} \|\Pi(p - \nu_i)\|^2 \leq 2 \sum_i \sum_{p \in G_i} \|\Pi(p - \mu_i)\|^2 + \|\Pi(\mu_i - \nu_i)\|^2.$$

From Lemma 21, we know that $\sum_i \sum_{p \in G_i} \|\Pi(p - \mu_i)\|^2 = O(n \log n \cdot k \sigma_{\max}^2)$. The guarantee of $\nu_i$ in Lemma 23 ensures $\sum_i \sum_{p \in G_i} \|\Pi(\mu_i - \nu_i)\|^2 = O\left(n \cdot \log^2\left(\frac{1}{\varepsilon w_{\min}}\right) \cdot k \sigma_{\max}^2 \log n\right)$.

Thus, $\sum_p \|\Pi(p - f(p))\|^2 = O\left(n \cdot \log^2\left(\frac{1}{\varepsilon w_{\min}}\right) \cdot k \sigma_{\max}^2 \log n\right)$.

Since all points in $F$ have an estimated that satisfies $\big|\hat{w}_q - |f^{-1}(q)|\big| \leq \frac{|f^{-1}(q)|}{2}$, we can apply Lemma 13: the solution computed by the above algorithm on the dataset $\Pi F$ with weights $\hat{w}_q$ has cost at most $O\left(n \cdot \log^2\left(\frac{1}{\varepsilon w_{\min}}\right) \cdot k \sigma_{\max}^2 \log n\right) + O(\mathrm{OPT}(\Pi P)) = O\left(n \cdot \log^2\left(\frac{1}{\varepsilon w_{\min}}\right) \cdot k \sigma_{\max}^2 \log n\right)$.

This concludes the proof of Lemma 20.

## F. Part 2: Improving iteratively the solution

Our global algorithm is described in Algorithm 5: first, we use Lemma 17 to reduce the diameter of the input; then, we compute a good initial solution using Lemma 20. Then, we implement privately Part 2 and Part 3 of Algorithm 3, using private mean estimation. We note that this algorithm, when assumptions (1) and (2) are satisfied, is exactly Algorithm 1 followed with Algorithm 2. Hence, Theorem 3 follows directly from the proof of Theorem 15.

---

**Algorithm 5** Cluster

1: **Input:** Server data $Q$, client datasets $P^1, ..., P^m$, and privacy parameters $\varepsilon, \delta$
2: Process the input to reduce the diameter to $\Delta$ using Lemma 17, with privacy parameter $\varepsilon/4$.
3: In one round of communication, compute a $O(\sqrt{d}\Delta \cdot \sigma(\varepsilon/4, \delta))$-almost $k$-PCA using Theorem 9.
4: **Part 1:** find initial centers $\nu_1^{(1)}, ..., \nu_k^{(1)}$ using Lemma 20, with privacy parameter $\varepsilon/4$
5: **Part 2:**

    a) Server sends $\nu_1^{(1)}, .., \nu_k^{(1)}$ to clients, and client $c$ computes $S_r^c := \{P_i \in P^c : \forall s, \|\hat{P}_i - \nu_r\| \leq \frac{1}{3}\|\hat{P}_i - \nu_s\|\}$.

    b) Server receives, for all cluster $r$, $\nu_r^{(2)} := \frac{1}{\sum_{\text{client } c} |S_r^c| + \mathrm{Lap}(T/\varepsilon)} \left(\sum_{\text{client } c} \sum_{P_i \in S_r^c} P_i + \mathcal{N}_d\left(0, \frac{2T^2 \Delta \log(2T/\delta)}{\varepsilon^2}\right)\right)$

6: **Part 3:** Repeat Lloyd's steps for $T$ steps, with privacy parameter $(\varepsilon/T, \delta/T)$:

    1. Server sends $\nu_1^{(t)}, .., \nu_k^{(t)}$ to clients, and client $c$ computes $S_r^c := \{P_i \in P^c : \forall s, \|\hat{P}_i - \nu_r\| \leq \|\hat{P}_i - \nu_s\|\}$.

    2. Server receives, for all cluster $r$, $\nu_r^{(t+1)} := \frac{1}{\sum_{\text{client } c} |S_r^c| + \mathrm{Lap}(T/\varepsilon)} \left(\sum_{\text{client } c} \sum_{P_i \in S_r^c} P_i + \mathcal{N}_d\left(0, \frac{2T^2 \Delta \log(2T/\delta)}{\varepsilon^2}\right)\right)$

---

Given the mapping of Fact 19, the main result of Awasthi & Sheffet (2012) is that step 2 of the algorithm computes centers that are very close to the $\mu_i$:[5]

---

[5]Note that the original theorem of Awasthi & Sheffet (2012) is stated slightly differently: however, their proof only requires Fact 18 and the matching provided by Fact 19, and we modified the statement to fit our purposes.

**Theorem 27** (Theorem 4.1 in Awasthi & Sheffet (2012))**.** *Suppose that the solution $\nu_1, ..., \nu_k$ is as in Fact 19, namely, for each $\mu_i$, it holds that $\|\mu_i - \nu_i\| \leq 6\Delta_i$. Denote $S_i = \{j : \forall r \neq i, \|\Pi P_j - \nu_i\| \leq \frac{1}{3}\|\Pi P_j - \nu_r\|\}$. Then, for every $i \in [k]$ it holds that*

$$\|\mu(S_i) - \mu_i\| = O\left(\frac{1}{c\sqrt{|G_i|}} \cdot \|P - C\|_2\right),$$

*where $c$ is the separation constant from Definition 2.*

Finally, the next result from Kumar & Kannan (2010) shows that the Lloyd's steps converge towards the true means:

**Theorem 28** (theorem 5.5 in Kumar & Kannan (2010))**.** *If, for all $i$ and a parameter $\gamma \leq ck/50$,*

$$\|\mu_i - \nu_i\| \leq \frac{\gamma\|P - C\|_2}{\sqrt{|G_i|}},$$

*then*

$$\|\mu_i - \mu(C(\nu_i))\| \leq \frac{\gamma\|P - C\|}{2\sqrt{|G_i|}},$$

*where $C(\nu_i)$ is the set of points closer to $\nu_i$ than to any other $\nu_j$.*

This allows us to conclude the accuracy proof of Theorem 15

*Proof of Theorem 15.* The algorithm is $(\varepsilon, \delta)$-DP: each of the 4 steps step – reducing the diameter, computing a PCA, finding a good initial solution and running $T$ Lloyd's steps – is $(\varepsilon/4, \delta/4)$-DP, and private composition concludes.

The first three steps require a total of $2 + 10\log\frac{4\log|Q|}{\varepsilon w_{\min}}$ many rounds of communication, the last one requires $T + \log\frac{\sigma_{\max}^2}{w_{\min}}$ rounds. This simplifies to $T + \zeta_2\log\frac{\sigma_{\max}\log|Q|}{\varepsilon w_{\min}}$, for some constant $\zeta_2$.

The first step reduces the diameter to $\Delta = O\left(\frac{k\log^2 n\sqrt{d}\sigma_{\max}}{\varepsilon w_{\min}}\right)$; therefore, Lemma 20 combined with Fact 19 ensures that $\nu_1^{(1)}, ..., \nu_k^{(1)}$ satisfies the condition of Theorem 27. In addition, Lemma 4.2 of Awasthi & Sheffet (2012) ensures that the size of each cluster $|S_r|$ is at least $\frac{|G_i|}{2}$ at every time step.

Therefore, the private noise $\frac{\mathcal{N}_d(\Delta^2\sigma^2(\varepsilon',\delta'))}{|S_r^c|}$ is bounded with high probability by $\eta := O\left(\frac{\Delta\sqrt{d}\sigma(\varepsilon/T,\delta/T)}{|S_r^c|}\right) = O\left(\frac{kdT\log^2 n\sigma_{\max}\cdot\sqrt{\ln(1/\delta)}}{n\varepsilon^2 w_{\min}^2}\right)$, which for and $n = \Omega\left(\frac{kdT\log^2 n\cdot\sqrt{\ln(1/\delta)}}{\varepsilon^2 w_{\min}^2}\right)$ is smaller than $\Delta_i = \frac{\sigma_{\max}}{\sqrt{w_i}}\min\left(\sqrt{k}\operatorname{polylog}(d/w_{\min}), d\right)$.

Hence, the conditions of Theorem 27 and Theorem 28 are still satisfied after adding noise, and the latter implies that the noisy Lloyd steps converge exponentially fast towards $B(\mu_i, \eta)$.

More precisely, it holds with probability $1 - 1/k^2$ that $\left\|\mu_i - \nu_i^{T+\log\frac{\sigma_{\max}^2}{w_{\min}}}\right\| = O\left(\frac{1}{c2^{T+\log\frac{\sigma_{\max}^2}{w_{\min}}}}\right) \cdot \frac{\|P-C\|_2}{\sqrt{|G_i|}} + \eta$.

From Lemma 11 ensures $\|P - C\| \leq O(\sqrt{n}\sigma_{\max})$. Since $|G_i| \geq nw_{\min}/2$, the first term is at most $O\left(\frac{1}{2^T}\right)$.

Therefore,

$$\left\|\mu_i - \nu_i^{T+\log\frac{\sigma_{\max}^2}{w_{\min}}}\right\| = O\left(\max\left(\frac{1}{2^T}, \frac{kdT\log^2 n\sigma_{\max}\sqrt{\ln(T/\delta)}}{n\varepsilon^2 w_{\min}^2}\right)\right).$$

$\square$

# G. Experiment Details

## G.1. Dataset details

**Mixture of Gaussians Datasets** We generate a mixture of Gaussians in the following way. We set the data dimension to $d = 100$ and we generate $k = 10$ mixtures by uniformly randomly sampling $k$ means $\{\mu_1, \ldots \mu_k\}$ from $[0, 1]^d$. Each

mixture has diagonal covariance matrix $\Sigma_i = 0.5 I_d$ and equal mixture weights $w_i = 1/k$. The server data is generated by combining samples from the true mixture distribution together with additional data sampled uniformly randomly from $[0, 1]^d$ representing related but out-of-distribution data. We sample 20 points from each mixture component, for a total of $20 \times k = 200$ in distribution points and sample an additional 100 uniform points. For Section 5.1 we simulate a cross-silo setting with 100 clients, with each client having 1000 datapoints sampled i.i.d from the Gaussian mixture. For Section 5.2 we simulate a cross-device setting with 1000, 2000 and 5000 clients, each client having 50 points i.i.d sampled from the Gaussian mixture distribution. The server data is identical in both cases.

**US Census Datasets**    We create individual datapoints coming from the ACSIncome task in folktables. Thus each datapoint consists of $d = 819$ binary features describing an individual in the census, including details such as employment type, sex, race etc. In order to create a realistic server dataset (of related but not not in-distribution data) we filter the client datasets to contain only individuals of a given employment type. The server then receives a small amount (20) of datapoints with the chosen employment type, and a larger amount (1000) of datapoints sampled i.i.d from the set of individuals with a different employment type. We do this for 3 different employment types, namely "Employee of a private not-for-profit, tax-exempt, or charitable organization", "Federal government employee" and "Self-employed in own not incorporated business, professional practice, or farm". These give us three different federated datasets, each with 51 clients, with total dataset sizes of 127491, 44720 and 98475 points respectively.

**Stack Overflow Datasets**    Each client in the dataset is a stackoverflow user, with the data of each user being the questions they posted. Each question also has a number of tags associated with it, describing the broad topic area under which the question falls. We first preprocess the user questions by embedding them using a pre-trained sentence embedding model (Reimers & Gurevych, 2019). Thus a user datapoint is now a $d = 384$ text embedding. Now we again wish to create a scenario where the server can receive related but out of distribution data. We follow a similar approach to the creation of the US census datasets. We select two tag topics and filter our clients to consist of only those users that have at least one question that was tagged with one of the selected topics. For those clients we retain only the questions tagged with one of the chosen topics. The server then receives 1000 randomly sampled questions with topic tags that do not overlap with the selected client tags as well as 20 questions with the selected tags, 10 of each one. For our experiments we use the following topic tag pairs to create clients [(machine-learning, math), (github, pdf), (facebook, hibernate), (plotting, cookies)]. These result in federated clustering problems with $[10394, 9237, 23266, 2720]$ clients respectively.

## G.2. Verifying our assumptions

On each of the datasets used in our data-point-level experiments we compute the radius of the dataset $\Delta$, shown in Table 1.

| Dataset | $\Delta$ |
|---|---|
| Gaussian Mixture (100 clients) | 10.57 |
| US Census (Not-for-profit Employees) | 2.65 |
| US Census (Federal Employees) | 2.65 |
| US Census (Self-Employed) | 2.65 |

*Table 1.* Radius of each dataset.

Assumption (1) requires $\Delta = O\left(\frac{k \log^2(n) \sigma_{\max} \sqrt{d}}{\varepsilon w_{\min}}\right)$. For the Gaussian mixture, $k = 10, d = 100, w_{\min} = 1/10, n = 10^6$ and $\sigma_{\max} = 0.5$: thus $\Delta$ clearly satisfies the condition.

For the US Census datasets, $k = 10, d = 819, n \in \{127491, 44720, 98475\}$. As we cannot estimate $\sigma_{\max}$ and $w_{\min}$(since the dataset is not Gaussian), we use an upper-bound $w_{\min} = 1$, and replace $\sigma_{\max}$ with a proxy based on the optimal $k$-means cost, $\sqrt{\text{OPT}/n}$: this is a priori a large upper-bound on the value of $\sigma_{\max}$, but it still gives an indication on the geometry of each cluster. As can be seen in Figure 1, Figure 8, the average optimal cost is about $3.5$ : thus, $\sqrt{\text{OPT}/n} \approx 1.87$, and we estimate $\frac{k \log^2(n) \sigma_{\max} \sqrt{d}}{\varepsilon w_{\min}} \approx \frac{10 \cdot \log^2(10^5) \cdot 0.005 \cdot \sqrt{819}}{0.5} \approx 123000$. This indicates that Condition (1) is satisfied as well for this dataset.

Assumption 2 requires that the size of the server data is not too large: $|Q| \leq \frac{\varepsilon n k \log(n) \sigma_{\max}^2}{\Delta^2}$. In the Gaussian case, we have $|Q| = 300$, and the right-hand side is about 29000.

In the US Census Dataset, we again upper-bound $\sigma_{\max}^2 = \frac{\text{OPT}}{n}$. In that case, the right-hand side is about 620000, while there are 1020 server point. Although our estimate of $\sigma_{\max}$ is only an upper-bound, this indicates that assumption (2) is also

satisfied.

### G.3. Baseline implementation details

**SpherePacking**    We implement the data independent initialization described in Su et al. (2017) as follows. We estimate the data radius $\Delta$ using the server dataset. We set $a = \Delta\sqrt{d}$, for $i = 1, \ldots, k$, we randomly sample a center $\nu_i$ in $[-\Delta, \Delta]^d$. If $\nu_i$ is at least distance $a$ from the corners of the hypercube $[-\Delta, \Delta]^d$ and at least distance $2a$ away from all previously sampled centers $\nu_1, \ldots, \nu_{i-1}$, then we keep it. If not we resample $\nu_i$. We allow 1000 attempts to sample $\nu_i$, if we succeed with sampling all $k$ centers then we call the given $a$ feasible. If not then $a$ is infeasible. We find the largest feasible $a$ by binary search and use the corresponding centers as the initialization.

### G.4. Adapting FedDP-KMeans to client-level privacy

As discussed in Section 5.2, moving to client-level DP changes the sensitivities of the algorithm steps that use client data. To calibrate the noise correctly we enforce the sensitivity of each step by clipping the quantities sent by each client to the server, prior to them being aggregated.

Concretely, suppose $v_j$ is a vector quantity owned by client $j$, and the server wishes to compute the aggregate $v = \sum_j v_j$. Then prior to aggregation the client vector is clipped to have maximum norm $B$ so that

$$\hat{v}_j = \begin{cases} \frac{B}{\|v_j\|} v_j, & \text{if } \|v_j\| > B \\ v_j, & \text{otherwise.} \end{cases}$$

The aggregate is then computed as $\hat{v} = \sum_j \hat{v}_j$. This query now has sensitivity $B$, and noise can be added accordingly. Each step of our algorithms can be expressed as such an aggregation over client statistics, the value of $B$ for each step becomes a hyperparameter of the algorithm.

We make one additional modification to Step 3 of FedDP-Init to make it better suited to the client-level DP setting. In Algorithm 1 during Step 3 the clients compute the sum $m_r^j$ and count $n_r^j$ of the vectors in each cluster $S_r^j$. Rather than send these to the server to be aggregated the client instead computes their cluster means locally as

$$u_r^j = \begin{cases} \frac{m_r^j}{n_r^j}, & \text{if } n_r^j > 0 \\ 0, & \text{otherwise,} \end{cases}$$

as well as a histogram counting how many non-empty clusters the client has:

$$c_r^j = \begin{cases} 1, & \text{if } n_r^j > 0 \\ 0, & \text{otherwise.} \end{cases}$$

The server then receives the noised aggregates $\widehat{u_r}$ and $\widehat{c_r}$ and computes the initial cluster centers as $\nu_r = \widehat{u_r}/\widehat{c_r}$. In other words we use a mean of the means estimate of the true cluster mean.

## H. Additional Experiments

### H.1. Setting hyperparameters of FedDP-KMeans

In this section we analyze the hyperparameter settings of FedDP-KMeans that produced the Pareto optimal results shown in the figures in Sections 5.1 and 5.2. These analyses give us some insights on the optimal ways to set the hyperparameters when using FedDP-KMeans in practice.

**Distributing the privacy budget**    The most important parameters to set are the values of epsilon in Parts 1-3 of Algorithm 1. Here we discuss how to set these.

Let $\varepsilon_1$, $\varepsilon_2$, $\varepsilon_{3G}$ and $\varepsilon_{3L}$ denote the epsilon we allow for part 1, part 2, the Gaussian query in part 3 and the Laplace query in part 3 respectively. We let $\varepsilon_{\text{init}} = \varepsilon_1 + \varepsilon_2 + \varepsilon_{3G} + \varepsilon_{3L}$. By strong composition the initialization will have a lower overall budget than $\varepsilon_{\text{init}}$, however, it serves as a useful proxy to the overall budget as we can think of what proportion of $\varepsilon_{\text{init}}$ we are assigning to each step.

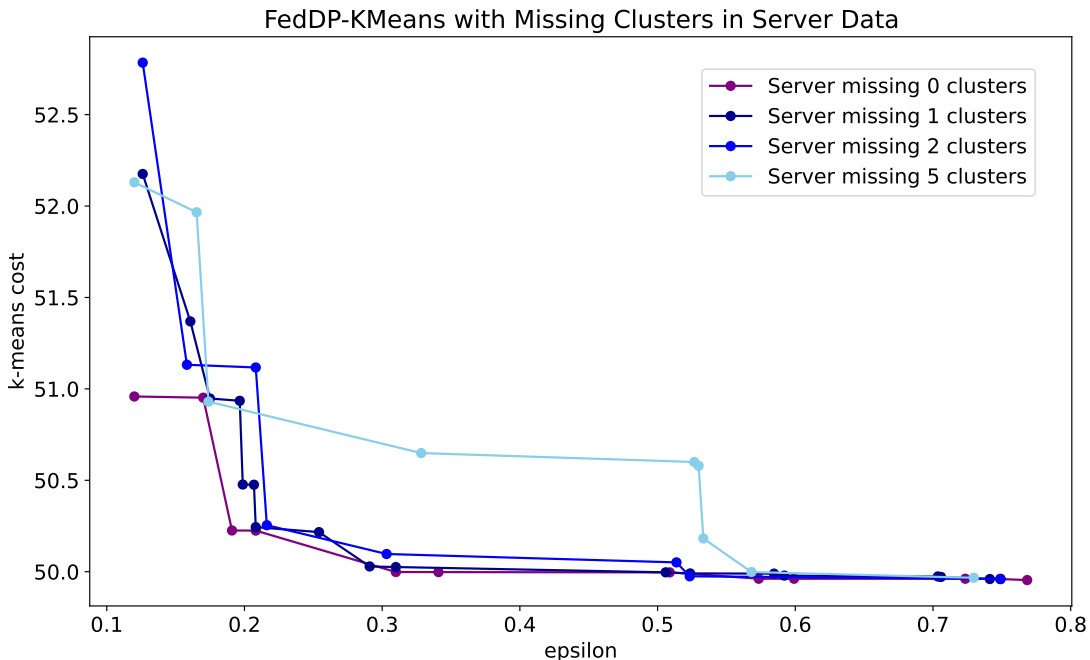

*Figure 3.* Mixture of Gaussians data with $k = 10$ clusters and 100 clients. Performance of FedDP-KMeans when the server is missing 0, 1, 2 and 5 of the 10 total clusters.

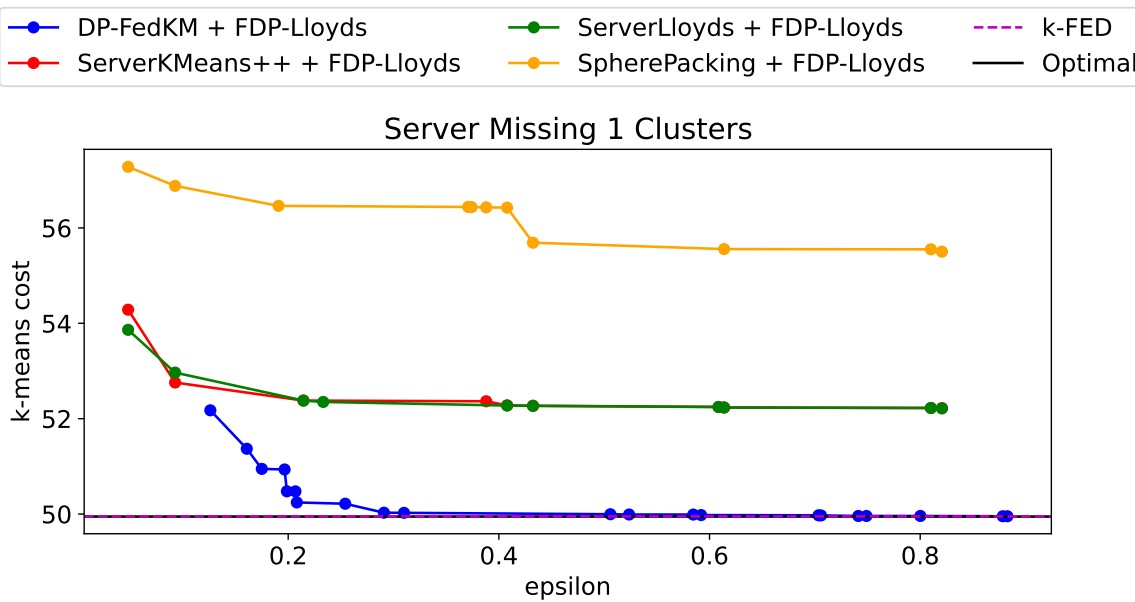

*Figure 4.* Mixture of Gaussians data with 100 clients. Server missing 1 of the $k = 10$ clusters.

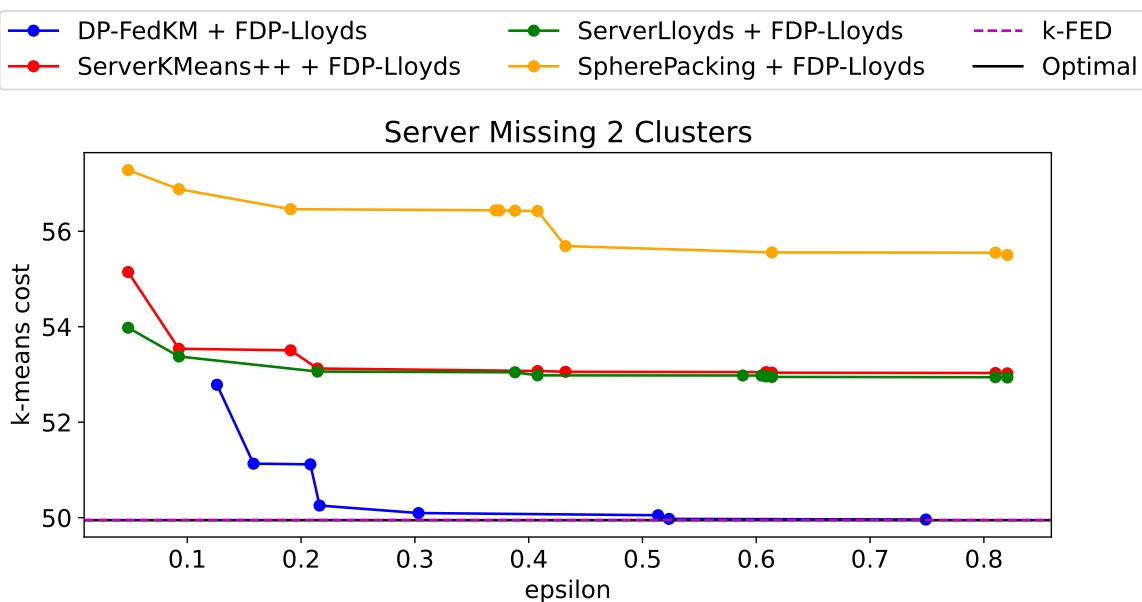

*Figure 5.* Mixture of Gaussians data with 100 clients. Server missing 2 of the $k = 10$ clusters.

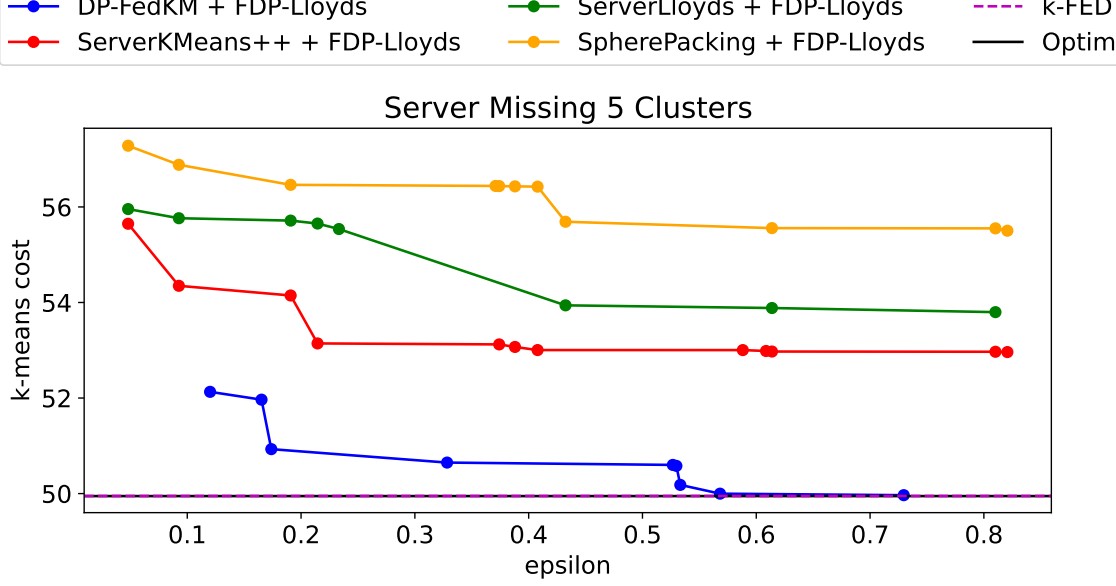

*Figure 6.* Mixture of Gaussians data with 100 clients. Server missing 5 of the $k = 10$ clusters.

Shown in Tables 2 and 3 are the values from our experiments. Specifically, for each dataset we take the mean across the Pareto optimal results that we plotted of the $\varepsilon$ values used for each step. We then express this as a fraction of $\varepsilon_{\text{init}}$. Loosely speaking, we interpret these values as answering "What fraction of our overall privacy budget should we assign to each step?"

The results paint a consistent picture when comparing values with the same unit-level of privacy with slight differences between the two levels. For datapoint level privacy, clearly the most important step in terms of assigning budget is to the Gaussian mechanism in Step 3 with the other steps being roughly even in term of importance. Therefore, as a rule of thumb we would recommend assigning budget using the following approximate proportions $[0.2, 0.2, 0.45, 0.15]$. For user level privacy we observe the same level of importance being placed on the Gaussian mechanism in Step 3 but additionally on the Gaussian mechanism in Step 1. Based on these results we would assign budget following approximate proportions $[0.35, 0.1, 0.45, 0.1]$. Clearly these are recommendations based only on the datasets we have experimented with and the optimal settings will vary from dataset to dataset, most notably based on the number of clients and the number of datapoints per client.

| Dataset | $\epsilon_1/\epsilon_{\text{init}}$ | $\epsilon_2/\epsilon_{\text{init}}$ | $\epsilon_{3G}/\epsilon_{\text{init}}$ | $\epsilon_{3L}/\epsilon_{\text{init}}$ |
|---|---|---|---|---|
| Gaussian Mixture (100 clients) | 0.18 | 0.23 | 0.43 | 0.17 |
| US Census (Not-for-profit Employees) | 0.24 | 0.17 | 0.41 | 0.18 |
| US Census (Federal Employees) | 0.15 | 0.16 | 0.52 | 0.17 |
| US Census (Self-Employed) | 0.20 | 0.23 | 0.47 | 0.10 |

*Table 2.* Amount of privacy budget, as a fraction of $\varepsilon_{\text{init}}$, that is assigned to each step of FedDP-Init. Results shown are the mean of the Pareto optimal results plotted for each of the data-point-level experiments in Figures 1, 8 and 9.

| Dataset | $\varepsilon_1/\varepsilon_{\text{init}}$ | $\varepsilon_2/\varepsilon_{\text{init}}$ | $\varepsilon_{3G}/\varepsilon_{\text{init}}$ | $\varepsilon_{3L}/\varepsilon_{\text{init}}$ |
|---|---|---|---|---|
| Gaussian Mixture (1000 clients) | 0.38 | 0.09 | 0.42 | 0.10 |
| Gaussian Mixture (2000 clients) | 0.43 | 0.10 | 0.36 | 0.11 |
| Gaussian Mixture (5000 clients) | 0.43 | 0.09 | 0.37 | 0.11 |
| Stack Overflow (facebook, hibernate) | 0.29 | 0.15 | 0.42 | 0.15 |
| Stack Overflow (github, pdf) | 0.37 | 0.12 | 0.40 | 0.10 |
| Stack Overflow (machine-learning, math) | 0.29 | 0.14 | 0.45 | 0.13 |
| Stack Overflow (plotting, cookies) | 0.33 | 0.11 | 0.47 | 0.09 |

*Table 3.* Amount of privacy budget, as a fraction of $\varepsilon_{\text{init}}$, that is assigned to each step of FedDP-Init. Results shown are the mean of the Pareto optimal results plotted for each of the client-level experiments in Figures 2, 10, 11, 12 and 13.

**Number of steps of FedDP-Lloyds** The other important parameter to set in FedDP-KMeans is the number of steps of FedDP-Lloyds to run following the initialization obtained by FedDP-Init. As discussed already, this comes with the inherent trade-off of number of iterations vs accuracy of each iteration. For a fixed overall budget, if we run many iterations, then each iteration will have a lower privacy budget and will therefore be noisier. Not only that, but in fact the question of whether we even want to run any iterations has the same trade-off. If we run no iterations of FedDP-Lloyds, then we use none of our privacy budget here, and we have more available for FedDP-Init. To investigate this we do the following: for each dataset we compute, for each number of steps $T$ of FedDP-Lloyds, the fraction of the Pareto optimal runs that used $T$ steps.

| Dataset | 0 steps | 1 step | 2 steps |
|---|---|---|---|
| Gaussian Mixture (100 clients) | 0.61 | 0.39 | 0 |
| US Census (Not-for-profit Employees) | 0.8 | 0.1 | 0.1 |
| US Census (Federal Employees) | 0.91 | 0.09 | 0 |
| US Census (Self-Employed) | 0.92 | 0.08 | 0 |

*Table 4.* Fraction of the Pareto optimal results that used a given number of steps of FedDP-Lloyds for the data-point-level experiments.

The results, shown in Tables 4 and 5, are interesting. In all but one dataset more than 80% of the optimal runs used no steps of FedDP-Lloyds, with many of the datasets being over 90%. The preference was to instead use all the budget for the

| Dataset | 0 steps | 1 step | 2 steps |
|---|---|---|---|
| Gaussian Mixture (1000 clients) | 0.86 | 0.11 | 0.04 |
| Gaussian Mixture (2000 clients) | 0.8 | 0.17 | 0.03 |
| Gaussian Mixture (5000 clients) | 0.81 | 0.1 | 0.1 |
| Stack Overflow (facebook, hibernate) | 1.0 | 0 | 0 |
| Stack Overflow (github, pdf) | 1.0 | 0 | 0 |
| Stack Overflow (machine-learning, math) | 0.94 | 0.06 | 0 |
| Stack Overflow (plotting, cookies) | 0.96 | 0.04 | 0 |

*Table 5.* Fraction of the Pareto optimal results that used a given number of steps of FedDP-Lloyds for the client-level experiments.

initialization. The reason for this is again the inherent trade-off between number of steps and accuracy of each step, with it clearly here being the case that fewer more accurate steps were better. One point to note here is that FedDP-Init essentially already has a step of Lloyds built into it, Step 3 is nearly identical to a Lloyds step but with points assigned by distance in the projected space. Running this step once and to a higher degree of accuracy tended to outperform using more steps. This in fact highlights the point made in our motivation, about the importance of finding an initialization that is already very good, and does not require many follow up steps of Lloyds.

### H.2. Missing clusters in the server data

In order for our theoretical guarantees to hold we required the assumption that the server data include at least one point sampled from each of the components of the Gaussian mixture and this assumption was reflected in the experimental setup of Sections 5.1 and 5.2. This is, however, not a requirement for FedDP-KMeans to run or work in practice.

To test this we run experiments in the setting that certain clusters are missing from the server data in our Gaussian mixtures setting. Specifically, the server data is constructed by sampling from only a subset of the $k = 10$ Gaussian components of the true distribution. Figure 3 shows the performance of FedDP-KMeans as we increase the number of clusters missing on the server. As we can see performance deteriorates modestly as the number of clusters missing from the server dataset increases. Figures 4-6 show that this also occurs in the other baselines that make use of the server data and that FedDP-KMeans is still the best performing method in this scenario.

### H.3. Choosing $k$ using Weighted Server Data

While in practice, $k$-means is often used with a value of $k$ determined by external factors, such as computational or memory demands, $k$ can also be chosen based on the data at hand. Existing methods can be incorporated into our setting quite simply, by using the method on the weighted and projected server dataset $\Pi Q$, with weights $w_q(\hat{\Pi} P)$. This dataset serves as a proxy for the client data and we can operate on it without incurring any additional privacy costs. We illustrate this using the popular elbow method. Concretely, we run lines 1-16 of Algorithm 1 using some large value $k'$, then we run line 17 for $k = 1, 2, 3, \ldots$ and plot the $k$-means costs of the resulting clusterings. This is shown in Figure 7. Clearly, the elbow of the curve occurs at $k = 10$ which is indeed the number of clusters in the true data distribution (we used the same Gaussian mixture dataset as in the original experiments).

## I. Additional figures

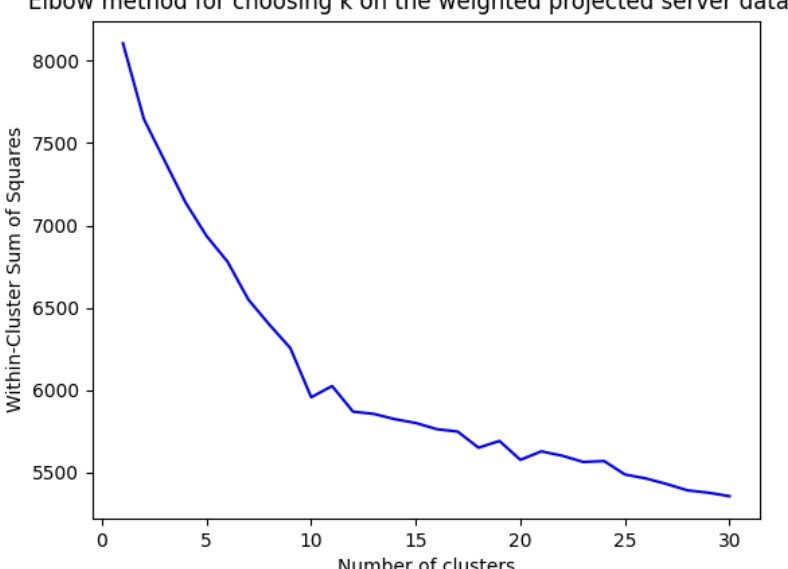

*Figure 7.* Plotting the Within-Cluster Sum of Squares (aka $k$-means cost), against the number of clusters, when clustering the weighted projected server data points. The true number of clusters in the data is $k = 10$, the prior steps of FedDP-Init were run with $k' = 20$. The "elbow" of the curve indeed occurs at $k = 10$.

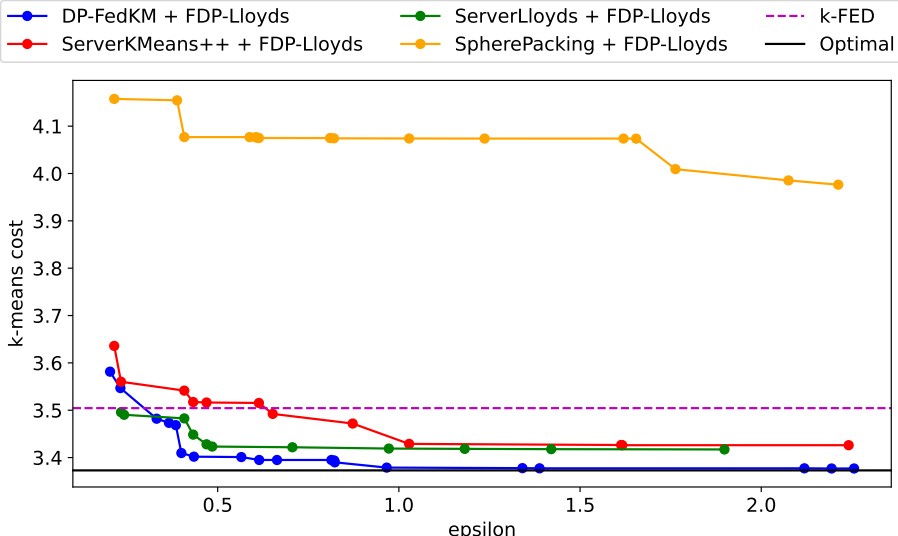

*Figure 8.* Results with data-point-level privacy on US census data. The 51 clients are US states, each client has the data of individuals with employment type "Employee of a private not-for-profit, tax-exempt, or charitable organization".

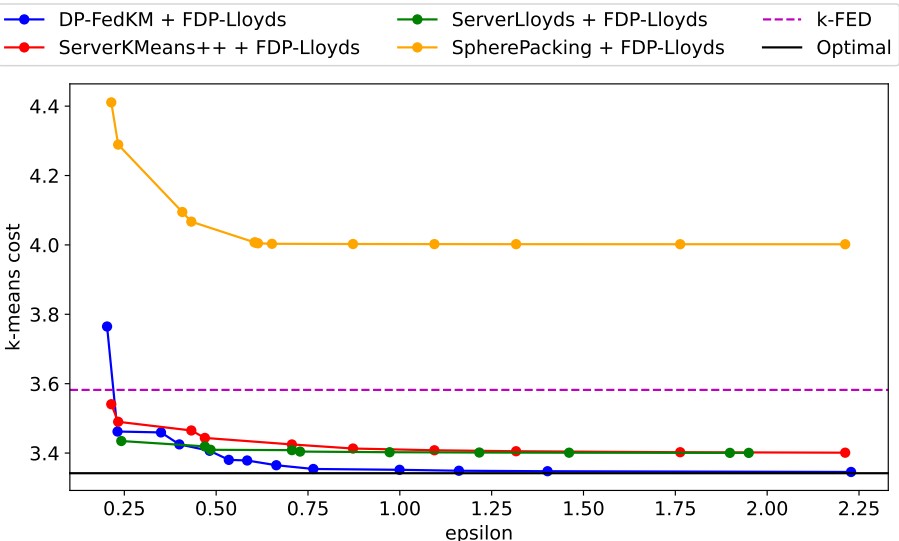

*Figure 9.* Results with data-point-level privacy on US census data. The 51 clients are US states, each client has the data of individuals with employment type "Self-employed in own not incorporated business, professional practice, or farm".

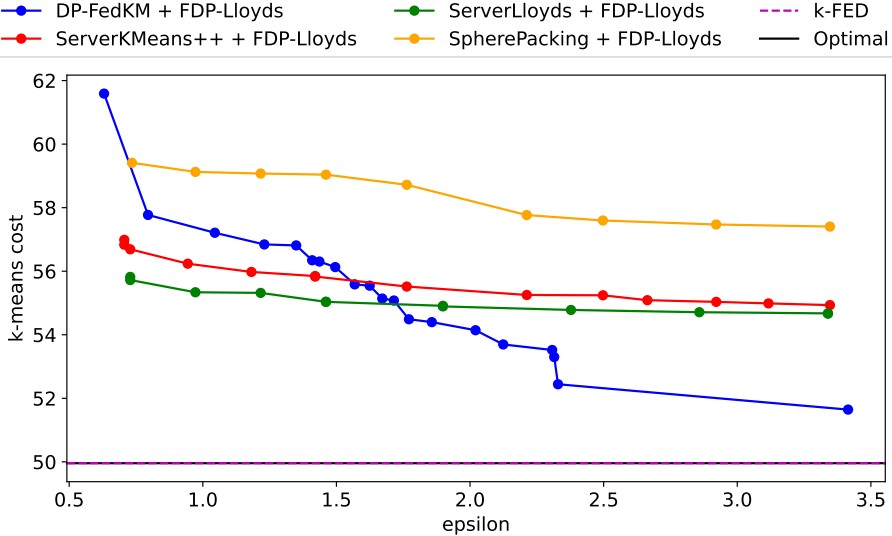

*Figure 10.* Results with client-level privacy on Synthetic mixture of Gaussians data with 1000 clients in total.

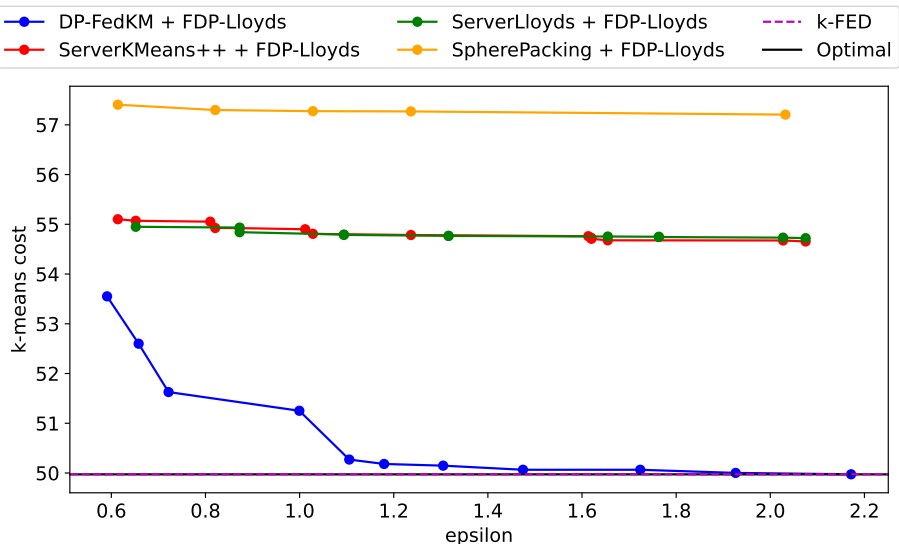

*Figure 11.* Results with client-level privacy on Synthetic mixture of Gaussians data with 5000 clients in total.

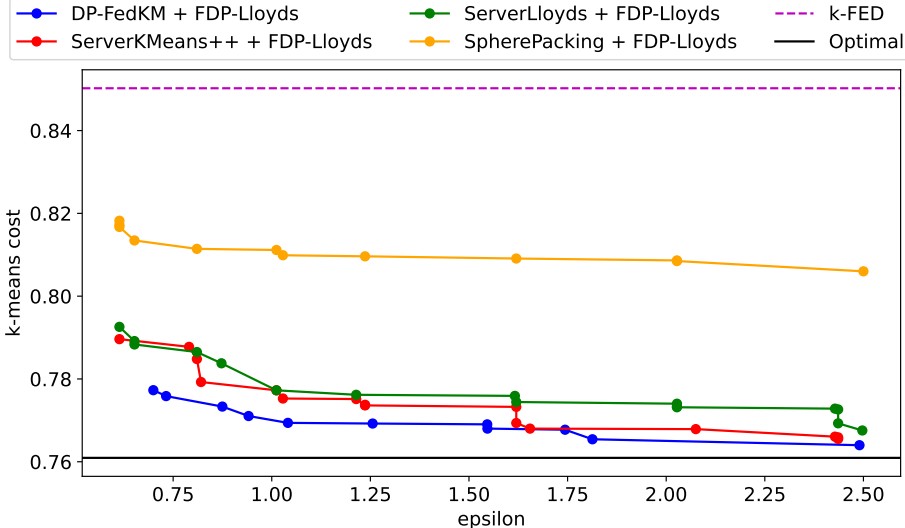

*Figure 12.* Results with client-level privacy on the stackoverflow dataset with 23266 clients with topic tags facebook and hibernate.

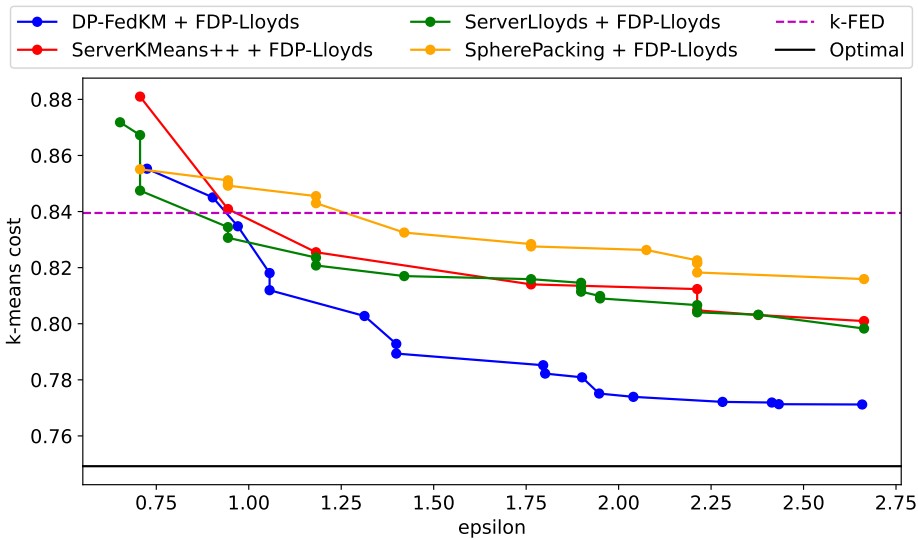

*Figure 13.* Results with client-level privacy on the stackoverflow dataset with 2720 clients with topic tags plotting and cookies.

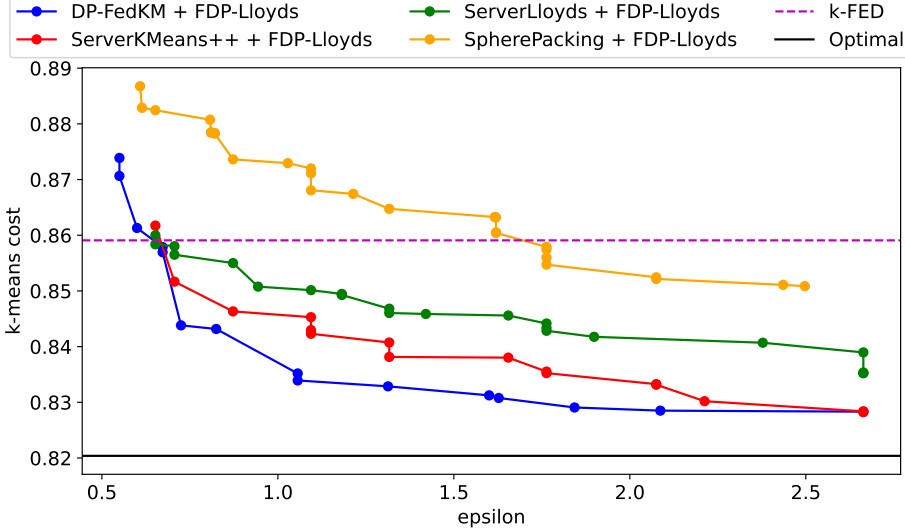

*Figure 14.* Results with client-level privacy on the stackoverflow dataset with 10394 clients with topic tags machine-learning and math.

