# OpenReview forum: "Differentially Private Federated $k$-Means Clustering with Server-Side Data"
_ICML.cc/2025/Conference — ICML 2025 poster_

### Official Review · Reviewer_DeqA · 2025-03-09

**Overall Recommendation:** 3

**Summary:**

This paper proposes a novel fully federated and differentially private k-means clustering algorithm (FedDP-KMeans). This method overcomes the problem that existing differentially private (DP) clustering methods require good clustering initialization by utilizing the data on the server side. Experiments have been conducted under the data-point-level privacy and client-level privacy settings to verify the effectiveness of this method.

**Claims And Evidence:**

Yes

**Essential References Not Discussed:**

The following highly related works have not been cited/discussed:
[1] Zhang Y, Chen H, Lin Z, et al. FedAC: An Adaptive Clustered Federated Learning Framework for Heterogeneous Data. arXiv preprint arXiv:2403.16460, 2024.
[2] Zhang Y, Zhang Y, Lu Y, et al. Asynchronous Federated Clustering with Unknown Number of Clusters. arXiv preprint arXiv:2412.20341, 2024.
[3] Ma Q, Xu Y, Xu H, et al. FedUC: A unified clustering approach for hierarchical federated learning. IEEE Transactions on Mobile Computing, 2024.
[4] Zhang Y, Chen H, Lin Z, et al. Lcfed: An efficient clustered federated learning framework for heterogeneous data. arXiv preprint arXiv:2501.01850, 2025.

**Ethical Review Concerns:**

No.

**Experimental Designs Or Analyses:**

Yes, I have checked. There is a lack of experiments for evaluating the clustering effect using external clustering evaluation indices such as the Fowlkes-Mallows Index (FMI), Adjusted Rand Index (ARI), and internal indices such as the Silhouette Coefficient (SC) and Calinski-Harabasz Index (CH). Ablation experiments are also lacking, as well as some parameter analysis experiments, such as the number of clients \(m\), privacy parameters \(\varepsilon\), etc.

**Methods And Evaluation Criteria:**

Yes

**Other Comments Or Suggestions:**

I do not have.

**Other Strengths And Weaknesses:**

Strengths：

This paper proposes a novel fully federated and differentially private k-means clustering algorithm (FedDP-KMeans). This method overcomes the problem that existing differentially private (DP) clustering methods require good clustering initialization by utilizing the data on the server side. Experiments have been conducted under the data-point-level privacy and client-level privacy settings to verify the effectiveness of this method.

Weaknesses：

1. This paper lacks a summary of the contributions. The authors should clearly list the core contribution points of this paper so that readers can clearly grasp the innovative value of the research.

2. The main contribution is the proposal of a new initialization method. However, a large part of the introduction section is spent on introducing the development and limitations of privacy techniques and federated learning. There is no detailed elaboration on the existing problems of the current initialization methods and the challenges they bring. Nor does it explain how this initialization method can solve the above problems.

**Questions For Authors:**

Since the method proposed in this paper is for clustering private data, how do the authors solve the problem of the unknown number of clusters \(k\) and the problem that k-means performs poorly on non-convex datasets?

And also see the weaknesses.

If the concerns can be well addressed, I will consider raise the score.

**Relation To Broader Scientific Literature:**

This paper proposes a novel fully federated and differentially private k-means clustering algorithm (FedDP-KMeans). Most of the existing federated clustering methods do not have such a high level of privacy protection, but only use a federated architecture to ensure that local original information is not directly transmitted.

**Theoretical Claims:**

Yes

---

> ### Author Rebuttal · Authors · 2025-03-29
>
> Thank you for your review and helpful comments. We have run additional experiments which can be found here https://anonymous.4open.science/r/FedDP-KMeans-Rebuttal-Figures-5B34/Rebuttal_Figures.pdf. We will reference this pdf when we address your specific concerns below.
>
> > lack of experiments for evaluating the clustering effect using external clustering evaluation indices
>
> We have run experiments evaluating with the suggested Fowlkes-Mallows and Adjusted Rand Indices, see FIgures 6-9 of the additional experiments. The results are in line with those reported with k-means cost.
>
> We would like to add that we believe that the metric we report, k-means cost, is fair and informative. All algorithms we compare are variants of k-means, so they all have the goal of minimizing the k-means objective.
>
> > Ablation experiments are also lacking, as well as some parameter analysis experiments, such as the number of clients (m), privacy parameters (\varepsilon)
>
> Thank you for the suggestion, we have run additional ablation experiments testing how our method performs when the server data is missing some of the clusters. See Section 1 of the additional experiments. The performance of all methods that use server data deteriorates modestly as the number of missing clusters grows. In all settings FedDP-KMeans is still the best performing method.
>
> Regarding number of clients, our current experiments do already cover a wide range of values of m. We have experiments with $m \in\lbrace 51,100,1000,2000,2720,5000,9237,10394,23266\rbrace$. Moreover, for the Gaussian mixture data we do in fact keep the distribution the same and vary $m \in\lbrace 1000, 2000, 5000\rbrace $ and discuss how the results change (Results paragraph on page 7). We hope this addresses your concerns.
>
> Regarding analysis of the privacy parameters we provide a detailed analysis of these in Appendix G.4 and our main experiments (e.g. Figures 1 and 2) analyze the changes in performance as privacy parameters vary. Do these points answer your specific concern?
>
> > highly related works have not been cited/discussed:
>
> Thank you for the references. It seems that [1], [3] and [4] are solving a related task of Clustered FL, where the goal is to cluster clients for better model training. [2] do propose a method for k-means clustering of the data, though their focus is on asynchronous and heterogeneous clients rather than privacy. We would be happy to include a discussion of these additional works in our Related Works section.
>
> > lacks a summary of the contributions.
>
> We will include a clearer summary of the main contributions of our work:
> - We propose a novel differentially private and federated initialization method that leverages small, out-of-distribution server-side data to generate high-quality initializations for federated $k$-means.
> - We introduce the first fully federated and differentially private $k$-means algorithm by combining this initialization with a simple DP federated Lloyd’s variant.
> - We provide theoretical guarantees showing exponential convergence to ground truth clusters under standard assumptions.
> - We conduct extensive empirical evaluations, demonstrating strong performance across data-point and user-level privacy settings on both synthetic and real federated data.
>
> > About the level details in the introduction:
>
> We aimed to make the work accessible to readers, who are not experts on privacy and FL, as these aspects are what make the problem of clustering much harder and unsolved. We’ll be happy to expand the discussion on initialization as well.
>
> > how do the authors solve the problem of the unknown number of clusters (k)
>
> Thank you for the question. In practice, k-means is often used with a value of k determined by external factors, such as computational or memory demands [Jain, 2010]. If k is meant to be chosen based on the data itself, existing methods can be incorporated into our setting quite simply, by using the method on the weighted and projected server dataset, $\Pi Q$ with weights $\hat{w_q(\Pi P)}$. This dataset serves as a proxy for the client data and we can operate on it without incurring any additional privacy costs. We illustrate this using the popular elbow method [Thorndike, 1953]. Concretely, we run lines 1-16 of Algorithm 1 using some large value $k’$, then we run line 17 for $k=1, 2, 3, \dots$ and plot the $k$-means costs of the resulting clusterings. This is shown in Figure 10 of the additional experiments. Clearly, the elbow of the curve occurs at $k=10$ which is indeed the number of clusters in the true data distribution (we used the same Gaussian mixture dataset as in the original experiments).
>
> > the problem that k-means performs poorly on non-convex datasets?
>
> Respectfully, we feel that this is not really a fair criticism of our method. Our contribution is a method for making k-means differentially private in a federated setting. It is not about trying to overcome principled shortcomings of the k-means objective.

---

### Official Review · Reviewer_octJ · 2025-03-10

**Overall Recommendation:** 4

**Summary:**

This submission proposes an $(\\epsilon, \\delta)$-differentially private algorithm for aligning a server-side k-means clustering with client-side data by private, federated computation. In particular, the authors propose an initialization procedure FedDP-Init, where clients compute an SVD on their data in a federated way, which is used by the server to compute a projection matrix and send it alongside with, intuitively, a projected k-means coreset of public server data, to the clients. Clients assign weights to the conceptual coreset points in a private fashion and return noisy means that the server can use to construct centers in the original space. Finally, a federated DP Lloyds algorithm is used to improve the solution.

The authors analyze their algorithm theoretically for well-separated Gaussian mixtures under the assumption that the server has at least one good sample from each component. Roughly speaking, they show that for each mean of a component, there is a center in the solution that has distance at most $O(\\sqrt{n \\sigma^2} + \\log(n)/(\\epsilon n)$.

For experiments, the authors consider point-level and client-level privacy on synthetic and real-world datasets. They benchmark FedDP-Init again two methods that use only server-side data, and an almost data-independent sphere packing, followed by FedDP-Lloyd. For small datasets with a few thousands of protected entities, FedDP-Init outperfoms the other solutions at around $\\epsilon \\approx 1$, while larger populations give the best solutions at $\\epsilon < 0.5$ already.

## update after rebuttal

The PC asks to add this section unconditionally. I didn't ask any questions in the rebuttal, so nothing has changed.

**Claims And Evidence:**

The claims are backed by theoretical results for a simple setting and experiments on synthetic and real-world data.

**Essential References Not Discussed:**

-

**Experimental Designs Or Analyses:**

The number of real-world data sets is a bit limited. Apart from the scale of the experiments, the approach is valid.

**Methods And Evaluation Criteria:**

Experiments plus a theoretical foundation are a valid approach.

**Other Comments Or Suggestions:**

None.

**Other Strengths And Weaknesses:**

In the non-private setting, it has been proven that a good initialization is key. The k-means++ initialization alone provides $O(\\log k)$ guarantees. Leveraging client data in addition to public server data to overcome misaligned distributions in a federated private setting is therefore a natural approach. Approximation and privacy are proven if the input comes from some Gaussian mixtures. The theoretical guarantees of the proposed approach fall short when some clusters are not represented in the server data at all even when they make up the majority of the client data, though. The experiments cover cases with unrelated noise, but do not explicitly cover the case that some regions of the input are not covered by the server data at all.

Overall, the theoretical results and the experiments suggest a significant value of center initialization with client data for either somewhat larger values of $\\epsilon$, or datasets with thousands or more of protected entities. However, the algorithm should be proven to be private on all input data sets, albeit without approximation guarantees for some inputs. Otherwise, privacy is ultimately not guaranteed.

**Questions For Authors:**

1. Could you confirm that the privacy is not proven if the input is not from a Gaussian mixture?

**Relation To Broader Scientific Literature:**

-

**Theoretical Claims:**

The proofs in the main part seem plausible.

---

> ### Author Rebuttal · Authors · 2025-03-29
>
> Thank you for your time and your review. We address your questions and comments below.
>
> > Could you confirm that the privacy is not proven if the input is not from a Gaussian mixture?
>
> This appears to be a misunderstanding that we find important to correct. Our algorithms (1 and 2) are always private, not just when the input comes from a Gaussian mixture. This is stated in Section 3.1 but we’ll add an explicit theorem to make this clearer.
>
> Technically, the $(\varepsilon, \delta)$-DP guarantee comes from the appropriately scaled noise that is added to the client statistics (e.g. lines 6, 15, 23 of Algorithm 1). Nowhere in the analysis of how noise is added to enforce privacy (section 3.1) do we require that the inputs come from a Gaussian mixture. Indeed our experiments are always run with $(\varepsilon, \delta)$ differential privacy, not just in the Gaussian mixture setting. The Gaussian mixture assumption is only required to theoretically prove accuracy and convergence. We also point out that, to define “accuracy” and “convergence”, assumptions such as the Gaussian mixture model are required.
>
> > The theoretical guarantees of the proposed approach fall short when some clusters are not represented in the server data at all even when they make up the majority of the client data, though. The experiments cover cases with unrelated noise, but do not explicitly cover the case that some regions of the input are not covered by the server data at all.
>
> For our theoretical guarantees you are of course correct. It is, however, not a requirement for the algorithm to work in practice. To test this we have now run additional experiments in exactly this setting that certain clusters are missing from the server dataset, the results can be found in Section 1 here: https://anonymous.4open.science/r/FedDP-KMeans-Rebuttal-Figures-5B34/Rebuttal_Figures.pdf. As seen in Figure 1, performance of FedDP-KMeans deteriorates modestly as the number of clusters missing from the server dataset increases. Figures 2-5 show that this also occurs in the other baselines that make use of the server data and that FedDP-KMeans is still the best performing method in this scenario. Thank you for the suggestion to consider this scenario in our experimental evaluation.

---

> > ### Comment · Reviewer_octJ · 2025-04-04
> >
> > Thank you the clarifying that the DP proof does not depend on the input being a Gaussian mixture, and the additional experiments!  I'd also suggest to change the section title "B.2. Differential Privacy for Gaussian Mixtures" and the phrasing of Theorem 8.

---

> > > ### Author Response · Authors · 2025-04-07
> > >
> > > Thank you for the additional helpful feedback!

---

### Official Review · Reviewer_daWe · 2025-03-13

**Overall Recommendation:** 3

**Summary:**

To adress the need of conducting clustering on distributed and private data, the authors proposed a private and disttubuted clustering framework. In detail, considering the performance of clustering highly relies on the initialization of the clustering center, the authors proposed ‘FedDP-Init’ that leverages a small-sclae server-side dataset and privatized client data statistics to provide a better initialization. The authors provide both theoretical and empirical evidences on their method works better than other baselines.

**Claims And Evidence:**

I think the claims are well supported.

**Essential References Not Discussed:**

N/A

**Experimental Designs Or Analyses:**

The experimental setting is rational, and the result seems sound.

**Methods And Evaluation Criteria:**

Considering  methods, the authors a better initilization for DP-Federated Clustering algorithms, which make sense sincei initilization is truly critical in clustering.
Considering evaluaton, they conduct experiments on both synthetic dataset and real-world dataset, which is relatively comprehensive.

**Other Comments Or Suggestions:**

N/A

**Other Strengths And Weaknesses:**

I would say the writing of this paper needs to be improved. Generally, I cannot grab a clear structure of Background-Motivation-Method-Contribution from the intrroduction. (‘background’ here doesn’t refers to the Section2, and I think the Section2 is more likely a Preliminary.)

**Questions For Authors:**

The main motivation of this work is the view that ‘initialization is important for clustering’, and that’s the reason why the author’s methods achieve such a clearly better expetimental results as reported in Fig1, 2. I’m curious that is there any other clustering method that is less sensitive to the initialization? And will your plugin improve those initialization-less-sensitive clustering method more marginally than Kmeans?

**Relation To Broader Scientific Literature:**

Generally, there are many works discussing potentials on DP-Fed-Clustering, despite its low accuracy due to excessive DP noise. This work provide an attempt on achieving better privacy-utility trade-off from the initilization directly.

**Theoretical Claims:**

The result of his convergence guarantee seems sound, as well as his privacy budget calculation. However I didn’t check each line in terms of all the authors’ proofs.

---

> ### Author Rebuttal · Authors · 2025-03-29
>
> Thank you for your time and for your review. We discuss your comments below.
>
> > I would say the writing of this paper needs to be improved. Generally, I cannot grab a clear structure of Background-Motivation-Method-Contribution from the intrroduction. (‘background’ here doesn’t refers to the Section2, and I think the Section2 is more likely a Preliminary.)
>
> Thank you for the feedback. We can rename Section 2 as Preliminary and rewrite the introduction so that the motivation is clearer. We will also include a more explicit summary of our contributions as follows:
> - We propose a novel differentially private and federated initialization method that leverages small, out-of-distribution server-side data to generate high-quality initializations for federated k-means.
> - We introduce the first fully federated and differentially private k-means algorithm by combining this initialization with a simple DP federated Lloyd’s variant.
> - We provide theoretical guarantees showing exponential convergence to ground truth clusters under standard assumptions.
> - We conduct extensive empirical evaluations, demonstrating strong performance across data-point and user-level privacy settings on both synthetic and real federated data.
>
> > I’m curious that is there any other clustering method that is less sensitive to the initialization? And will your plugin improve those initialization-less-sensitive clustering method more marginally than Kmeans?
>
> Of course. For example, there’s methods that do not need any initialization, such as single-linkage clustering (Gower,, Ross 1969), or that converge to the same solution, regardless of the initialization, such as convex clustering (Pelckmans et al 2005). But k-means is used much more in practice, because it has better properties.

---

### Official Review · Reviewer_ZG9J · 2025-03-14

**Overall Recommendation:** 4

**Summary:**

This paper proposes a k-means clustering algorithm in the federated learning model under differential privacy. The chief difficulty with such a setup is the seeding algorithm, since many non-private algorithms would be too slow in the federated learning model, and possibly not robust to the noise added by privacy. The primary contribution is a seeding algorithm which works using PCA to project into a lower dimension, then asking clients to add lower-dimensional noise to their projected vectors. Then, private Lloyd's algorithm is able to be run as normal. The utility of the approach is demonstrated through experiments and theoretically.

**Claims And Evidence:**

Yes.

**Essential References Not Discussed:**

I cannot think of essential references not discussed.

**Experimental Designs Or Analyses:**

I did not closely check the experiments for correctness.

**Methods And Evaluation Criteria:**

Yes.

**Other Comments Or Suggestions:**

No further comments.

**Other Strengths And Weaknesses:**

The algorithm is versatile, as it fits into common federated learning setups including those involving secure aggregation. It also can provide flexible privacy guarantees, including user-level and item-level differential privacy. The experiments are designed well, and there is a study on how to choose epsilon and other hyperparameters, which adds to the practical demonstration of the algorithm.

One potential drawback is in the requirement that the server have a representative datapoint for each cluster on hand. This assumption may be too strong, since often the purpose of running k-means clustering is to identify clusters.

**Questions For Authors:**

I don't fully understand the role of the server's private data Q. What would happen if the server simply took in each user's projected, noisy points and attempted to form seeds based on this? Can the assumption that Q need to contain a point from each cluster beforehand be relaxed at all?

**Relation To Broader Scientific Literature:**

This paper fits into private federated learning. Federated learning is an enormous field of research, and many works study how to apply privacy, which is usually applied the moment the data leaves the user's phone.

**Theoretical Claims:**

I did not closely check the proofs for correctness.

---

> ### Author Rebuttal · Authors · 2025-03-29
>
> Thank you for your time and for your review. We discuss your feedback below.
>
> > One potential drawback is in the requirement that the server have a representative datapoint for each cluster on hand. This assumption may be too strong, since often the purpose of running k-means clustering is to identify clusters.
>
> We would like to clarify one point: the server holds data points, but does not have any a priori information on the clustering on those points. In addition, the requirement that there is at least one point per cluster is only required for the theoretical contributions (where we want to identify ground truth GMM components). In practice, the algorithm performs well even without this assumption, see our answer below.
>
> > What would happen if the server simply took in each user's projected, noisy points and attempted to form seeds based on this?
>
> The issue with this is that directly sharing the projected user data points themselves with the server, while still having meaningful DP guarantees, would require so much noise as to destroy any signal in the points themselves. To give a concrete illustration, for the Gaussian mixtures data in the easier data-point level privacy setting with $\varepsilon = 2$, we would be forced to add iid 0 mean Gaussian noise with standard deviation $\approx35$ to each dimension of each projected user point before sending it to the server. So the expected norm of the noise vector to be added to each point is at least $35 \sqrt{10} \approx 105$. For reference the norms of the projected data points are mostly in the range (5, 7).
>
> > Can the assumption that Q need to contain a point from each cluster beforehand be relaxed at all?
>
> The requirement that the server holds a point from each cluster is only needed to prove our theoretical guarantees on clustering accuracy and convergence in the Gaussian mixtures setting. It is not strictly required for the algorithm to run or work in practice. We have now run additional experiments to test the method’s performance in the scenario where some clusters are completely missing in the server data, Q. The results can be found in Section 1 here: https://anonymous.4open.science/r/FedDP-KMeans-Rebuttal-Figures-5B34/Rebuttal_Figures.pdf. As seen in Figure 1, the performance of FedDP-KMeans deteriorates modestly as the number of clusters missing from the server dataset increases. Figures 2-5 show that this also occurs in the other baselines that make use of the server data and that FedDP-KMeans is still the best performing method in this scenario.

---

> > ### Comment · Reviewer_ZG9J · 2025-04-04
> >
> > The new plots look interesting! It looks like the clustering algorithm succeeds in finding the missing clusters when epsilon is high enough? Can you explain intuitively what is happening in the algorithm to cause this behavior?

---

> > > ### Author Response · Authors · 2025-04-07
> > >
> > > Indeed, your observation is correct: for sufficiently large $\varepsilon$, we can still match the performance of the optimal baseline.
> > >
> > > Intuitively, this likely occurs because some server data points—either from the OOD uniform distribution or outliers from the present Gaussians—are “close enough” (although not generated from the missing gaussian) in the projected space to receive non-negligible weights from client data associated with the missing Gaussians. These points then influence the server clustering, resulting in a projected center that somewhat approximates the true projected mean. This center helps locate the missing Gaussian in the client data, yielding a good initialization by the end of step 3.
> > >
> > > A larger $\varepsilon$ is required because the “close enough” points provide a weaker signal than true samples from the missing Gaussian. Since these points are fewer and further away, preserving their influence requires more accurate estimation of projection directions and server point weights—hence the need for a higher privacy budget.
> > >
> > > As a side note, this behavior may not generalize across all server data distributions. For example, if the server data consisted solely of very well-separated Gaussians, then it might be that there are no points “close enough” to the missing Gaussians. In that case, the projected server data would only reflect the present clusters and our initialization would likely fail to locate the missing Gaussians. As a result, FedDP-Lloyds might require more iterations than a typical privacy budget permits. However, such a scenario—completely non-overlapping clusters on the server—seems unrealistic in most practical settings.

---

### Decision · Program_Chairs · 2025-05-01

**Decision:**

Accept (poster)

**Comment:**

The paper proposes a new k-means clustering algorithm that is both "fully differentially private" and "fully federated", and it outperforms existing approaches that aim to achieve both goals simultaneously. While the new algorithm does not necessarily outperform prior state-of-the-art methods that were "only differentially private" or "only federated", it offers a noticeable improvement when both goals are pursued together. All the reviewers agree that this is an interesting paper that would make a solid contribution to ICML.